

# Cancelling mod-2 anomalies by Green-Schwarz mechanism with $B_{\mu\nu}$

**Shota Saito and Yuji Tachikawa**

Kavli Institute for the Physics and Mathematics of the Universe (WPI),
University of Tokyo, Kashiwa, Chiba 277-8583, Japan

## Abstract

We study if and when mod-2 anomalies can be canceled by the Green-Schwarz mechanism with the introduction of an antisymmetric tensor field $B_{\mu\nu}$. As explicit examples, we examine $SU(2)$ and more general $Sp(n)$ gauge theories in four and eight dimensions. We find that the mod-2 anomalies of 8d $\mathcal{N}=1$ $Sp(n)$ gauge theory can be canceled, as expected from it having a string theory realization, while the mod-2 Witten anomaly of 4d $SU(2)$ and $Sp(n)$ gauge theory cannot be canceled in this manner.

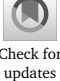

# 1 Introduction

It is well-known that the perturbative anomaly of a $d$-dimensional field theory, when its anomaly polynomial $I_{d+2}$ has a factorized form, can be canceled by an introduction of an antisymmetric tensor field. More precisely, suppose $I_{d+2} = X_{p+1}Y_{d-p+1}$, where the subscripts denote the degrees of the differential forms. We then introduce an antisymmetric tensor field with gauge-invariant field strength $H_p$ such that the relations

$$dH_p = X_{p+1}, \qquad d * H_p = Y_{d-p+1}, \tag{1}$$

are satisfied. Such an antisymmetric tensor field has the anomaly $-X_{p+1}Y_{d-p+1}$, which cancels the perturbative anomaly of the original theory.

This is the Green-Schwarz mechanism, and had already played a crucial part in establishing the existence of the ten-dimensional Type I and heterotic string theories [16]. In that case, we have a factorization of the form $I_{12} = X_4 Y_8$, and we introduce a field $H_3 = dB_2 + \dots$ satisfying

$$dH_3 = X_4, \qquad d * H_3 = Y_8. \tag{2}$$

The same mechanism can cancel a $U(1)G^2$ anomaly in four dimensions, since $I_{d+2} \propto F_{U(1)} \operatorname{tr} F_G^2$ in this case [34]. We can use $H_3$ as before, but using $H_1 = *H_3$ is also instructive in four dimensions. Then we have

$$dH_1 \propto F_{U(1)}, \qquad d * H_1 \propto \operatorname{tr} F_G^2. \tag{3}$$

This means that $H_1$ is the gauge-invariant derivative of a scalar $\phi$, i.e. $H_1 = d\phi + A_{U(1)}$. Furthermore, this scalar $\phi$ has an axion coupling $\propto \phi \operatorname{tr} F_G^2$. Therefore, $\phi$ eats $A_{U(1)}$ and both become massive, and the perturbative anomaly is gone.

All in all, we can say that a satisfactory theory of perturbative Green-Schwarz anomaly cancellation was already given in the mid 1980s. However, a theory with vanishing perturbative anomaly may still have a global anomaly, as first pointed out by Witten [33] in the case of $SU(2)$ gauge theories in four dimensions. This means that the perturbative Green-Schwarz anomaly cancellation is not enough. Even when the original fermion spectrum does not have a perturbative anomaly, it can have a global anomaly, which needs to be canceled by some mechanism if we expect or hope such a system to be consistent.

One example is the 8d $\mathcal{N}{=}1$ $Sp(n)$ gauge theory, which is known to have a string theory realization [35], whose fermion part has mod-2 anomalies. The anomaly cancellation of this system was studied in [14,23] with some partial success, but no complete understanding has been obtained so far.[1]

As these anomalies are discrete, one can try to cancel them by introducing some topological degrees of freedom, by a topological analogue of the Green-Schwarz mechanism. This was the approach advocated in [14] and pursued further e.g. in [30].

Another direction, which we will follow in this paper, is to carefully treat the antisymmetric tensor fields not just at the level of differential forms but also keeping track of their subtler topological information, so that we can examine the global anomalies they carry. This was the approach taken in [9,23]. In particular, in [23] it was shown that a part of the global mod-2 anomalies of the 8d $\mathcal{N}{=}1$ $Sp(n)$ gauge theory can be canceled in this manner by the introduction of the 3-form field $H_3$, but an uncanceled global anomaly remained. One of the main results of this paper is to show how this remaining global anomaly can be canceled. This will be done by combining the classifying space $BSp(n)$ of the gauge field and the classifying space $K(\mathbb{Z}, 3)$ of the $H$ field in a nontrivial fibration,

$$K(\mathbb{Z}, 3) \to Q \to BSp(n), \tag{4}$$

reflecting the condition $dH \propto \operatorname{tr} F^2$, and studying the anomaly by considering the bordism group of $Q$.[2]

One is then naturally led to the question whether the Witten anomaly of 4d $SU(2)$ and $Sp(n)$ gauge theories can be similarly canceled or not by the introduction of $H_3$, or equivalently by the introduction of an axion field $\phi$, as explained above.[3] The answer turns out to be negative, but the reasonings are quite subtle unless viewed from a particular perspective, as will be explained in detail below.

The organization of this paper is as follows. In Sec. 2, we recall the general bordism framework to understand the anomalies, and discuss how it can be used to formulate the anomaly cancellation condition, including both the perturbative part and the global part, after the introduction of an antisymmetric tensor field $H_p$. We also provide a detailed comparison to the previous analyses in the literature.

In Sec. 3, we study whether the 4d Witten anomaly of $SU(2)$ and $Sp(n)$ gauge theory can be canceled by an introduction of the field $H_3 = dB_2 + \ldots$, or equivalently the introduction of an axion $\phi$, using the general strategy established in Sec. 2. The answer turns out to be the negative, as can be seen from a relatively straightforward homotopy theory argument.

In Sec. 4, we go on to study the mod-2 anomaly cancellation of 8d $Sp(n)$ gauge theory, again by an introduction of the field $H_3 = dB_2 + \ldots$. We discuss two types of the modified Bianchi identify for $H_3$, namely $dH \propto \operatorname{tr} F^2$ and $dH \propto \operatorname{tr} F^2 - \operatorname{tr} R^2$. In the former case $dH \propto \operatorname{tr} F^2$, we will see that all mod-2 gauge anomalies can be canceled. In the latter case $dH \propto \operatorname{tr} F^2 - \operatorname{tr} R^2$, which is the case that actually arises in string theory, we find that not all mod-2 anomalies can be canceled. It turns out, however, that the actual massless fermion

---

[1]This 8d theory, originally introduced in [35], is a $T^2$ compactification of the $so(32)$ superstring 'without vector structure', and therefore has both the Type I and the heterotic realization. The most general analysis of global anomalies of Type I theories available as of now is [13], which unfortunately does not discuss the case 'without vector structure'. On the heterotic side, there is a general analysis showing the global anomaly cancellation [31], which unfortunately relies on a conjectural relation between topological modular forms and the classification of 2d minimally supersymmetric quantum field theories. The references [14,23] just cited are where more direct analyses in 8d were performed, and the present paper is a continuation of this more direct approach.

[2]This was the approach used to study global anomalies in six dimensions in [22].

[3]This question might sound even more attractive, if the reader knows the results of [6,7] that the mod-2 global Witten anomaly of $SU(2)$ simply becomes a perturbative $U(1)SU(2)^2$ anomaly in terms of a larger $U(2)$ symmetry without a discrete global part. In this case it is very natural to cancel this perturbative anomaly by the mechanism explained in (3). The issue then is whether any global anomalies remain after this operation.

spectrum provided by the string theory compactification is such that the resulting total mod-2 anomaly is cancellable, which can be considered as another string theory mini-miracle.

Then in Sec. 5, we provide algebraic topology computations required for the analysis of Sec. 4. In particular, we use Leray-Serre, Atiyah-Hirzebruch and Adams spectral sequences. We end our paper in Sec. 6 by giving an outlook of further possible directions of research.

We also have two Appendices. In Appendix A, we reach the conclusion of Sec. 3 in a more roundabout way, again by using Adams and Atiyah-Hirzebruch spectral sequences. And in Appendix B, we review the computation of $\pi_4(S^3)$ by using the Leray-Serre spectral sequence, as our algebraic-topological considerations in Sec. 3 and Sec. 5.2 are a direct generalization of this classic calculation. Finally in Appendix C, we give a generalization of our argument in Sec 3 to show that it is impossible to cancel traditional global anomalies characterized by $\pi_d(G)$ via the Green-Schwarz mechanism, using either continuous or discrete $p$-form fields.

## 2 Generalities

### 2.1 Global anomalies of fermions

In modern understanding, the anomaly of a $d$-dimensional theory defined on spin manifolds with symmetry group $G$ is captured by $(I_\mathbb{Z}\Omega^{\mathrm{spin}})^{d+2}(BG)$, the Anderson dual of the spin bordism group of the classifying space $BG$ of $G$, as was first formulated mathematically in [12]; for a detailed exposition to physicists, see e.g. [15, 21]. In this paper, we concentrate on the global anomalies, which correspond to the torsion part of $(I_\mathbb{Z}\Omega^{\mathrm{spin}})^{d+2}(BG)$, which is the Pontryagin dual to the torsion part of $\Omega_{d+1}^{\mathrm{spin}}(BG)$, the spin bordism group of $BG$.

Let $P \to M$ a $G$-bundle over a $(d+1)$-dimensional spin manifold $M$, giving rise to such a torsion element. Then a fermion in $d$ dimensions has a global anomaly detected by the bordism class $[P \to M] \in \Omega_{d+1}^{\mathrm{spin}}(BG)$ if the corresponding eta invariant on $M$ coupled to the $G$-bundle $P$ is nonzero.

For example, it is known that $\Omega_5^{\mathrm{spin}}(BSU(2)) = \mathbb{Z}_2$. A generator is obtained by taking a trivial $SU(2)$ bundle over $S^4 \times [0, 1]$, and connecting the two ends by a nontrivial $SU(2)$ gauge transformation $g : S^4 \to SU(2)$ given by the nontrivial element of $\pi_4(S^3) = \mathbb{Z}_2$. Another example in the same bordism class can be obtained by taking a five-dimensional sphere $S^5$, separating them into southern and northern hemispheres, considering trivial $SU(2)$ bundles over each hemisphere, and gluing them via the same gauge transformation $g : S^4 \to SU(2)$.

The exponentiated eta invariant of a fermion in the fundamental representation on this background is known to be $-1$, showing the fact that the fermion partition function on $S^4$ changes sign under such a gauge transformation $g$. This underlies the original global anomaly of Witten, found in [33].

The same bordism class is also known to be given by $S^4$ with an $SU(2)$ bundle with instanton number one, times $S^1$ with periodic spin structure. In such a product space of the form $(P' \to M') \times S^1$ with periodic spin structure around $S^1$, the exponentiated eta invariant of a fermion in representation $R$ is given by

$$(-1)^{\mathrm{ind}_{M'} D(R)}, \tag{5}$$

where $\mathrm{ind}_{M'} D(R)$ is the index of the Dirac operator $D(R)$ in the representation $R$ on $M'$ coupled to the gauge field $P'$. From this formula we can indeed check that the exponentiated eta invariant of the Dirac operator in the fundamental representation of $SU(2)$ on this background is $-1$.

These are the two main constructions of bordism classes which can detect global anomalies:

- One is to take an element $g : S^d \to G$ and use this to construct a $G$ bundle over $S^d \times S^1$ or $S^{d+1}$, by gluing the two ends of $S^d \times [0,1]$ or two boundaries of northern and southern hemispheres of $S^{d+1}$. They give rise to the same bordism class, and define a homomorphism

$$\pi_d(G) \to \Omega^{\mathrm{spin}}_{d+1}(BG). \tag{6}$$

- Another is to take a $G$-bundle on some manifold $M_d$, and consider its product with an $S^1$ with periodic spin structure. This gives a homomorphism

$$\Omega^{\mathrm{spin}}_d(BG) \to \Omega^{\mathrm{spin}}_{d+1}(BG). \tag{7}$$

These two homomorphisms (6) and (7) are neither surjective nor injective in general, but are often useful in constructing explicit bordism classes on the right hand side. The non-triviality of the homomorphism (6) has received some attention in the last few years, since its left hand side $\pi_d(G)$ was often regarded in the past as the main of the global anomaly, while the right hand side $\Omega^{\mathrm{spin}}_{d+1}(BG)$ was used more prominently as what detected the global anomalies in the more modern framework. For more on this interesting issue, see [8,22].

## 2.2 Introduction of the antisymmetric tensor field

Let us consider an antisymmetric tensor field $H_p$ with its gauge potential $B_{p-1}$. In isolation, the field strength $H_p$ is closed, $dH_p = 0$, and its topological data is captured by a class in $H^p(M;\mathbb{Z})$, or equivalently by the corresponding classifying map $M \to K(\mathbb{Z}, p)$. Here $K(A, n)$ is the Eilenberg-Mac Lane space, such that $\pi_n(K(A, n)) = A$ and $\pi_{m \neq n}(K(A, n)) = 0$.

We can modify its Bianchi identity so that the field strength $H_p$ is no longer closed. At the level of differential forms, this is done by choosing a characteristic class $X_{p+1}$ and demanding

$$dH_p = X_{p+1}. \tag{8}$$

For simplicity of presentation, let us assume until Sec. 2.4 that $X_{p+1}$ is solely constructed from the $G$-gauge field, i.e. an element of $H^{p+1}(BG;\mathbb{R})$.

To make it more precise, we suppose that the characteristic class $X_{p+1}$ is given by an element $X \in H^{p+1}(BG;\mathbb{Z})$, or equivalently by a map

$$X : BG \to K(\mathbb{Z}, p+1). \tag{9}$$

Then, we want to consider maps $f : M \to BG$ from the spacetime $M$ with the condition that $X \circ f$ is trivial homotopically, so that $f^*(X) \in H^{p+1}(M;\mathbb{Z})$ is the zero class.

Such a situation is answered nicely by the concept of the homotopy fiber of a map, see e.g. [17, Sec. 4.3]. Namely, given a map $X : A \to B$ between spaces, there is a homotopically unique space $F$ and a map $p : F \to A$ fitting in the sequence

$$F \xrightarrow{p} A \xrightarrow{X} B, \tag{10}$$

such that $f : M \to A$ satisfies $X \circ f$ being null if and only if $f : M \to A$ lifts to $f : M \to F$. Such an $F$ is known as the homotopy fiber of the map $X$. It is known that such a sequence can be extended to the left by taking the loop spaces:

$$\Omega B \to F \xrightarrow{p} A, \tag{11}$$

where the loop space $\Omega B$ of $B$ is now the homotopy fiber of $p : F \to A$.

Applying this construction to our situation, we find that the classifying space of the $G$-gauge field together with the antisymmetric tensor field is given by the homotopy fiber $Q$ of the map (9), i.e. the space fitting in the fiber sequence

$$Q \xrightarrow{p} BG \xrightarrow{X} K(\mathbb{Z}, p+1), \tag{12}$$

which in turn has the structure

$$K(\mathbb{Z}, p) \to Q \xrightarrow{p} BG, \tag{13}$$

where we used $\Omega K(\mathbb{Z}, p+1) = K(\mathbb{Z}, p)$. The equation (12) means that $Q$ is a universal space obtained from $BG$ by demanding that the characteristic class $X$ vanishes, and the equation (13) means that $Q$ obtained in this manner is a fibration of $K(\mathbb{Z}, p)$ over $BG$, i.e. by combining the degrees of freedom from the antisymmetric tensor field $H_p$ over the topological possibilities of the $G$ gauge field.

## 2.3 Condition for the anomaly cancellation

Suppose now that a theory defined on spin manifolds with symmetry group $G$ has an anomaly characterized by an element $I \in (I_{\mathbb{Z}}\Omega^{\mathrm{spin}})^{d+2}(BG)$. After adding the antisymmetric tensor field $H_p$ with its Bianchi identity specified by (8), the configuration space is changed from $BG$ to $Q$. Therefore, the anomaly after the introduction of $H_p$ is given by pulling back $I$ from $BG$ to $Q$ by the map $p : Q \to BG$ appearing in (12) and (13). Therefore the anomaly is canceled if and only if

$$p^*(I) = 0 \in (I_{\mathbb{Z}}\Omega^{\mathrm{spin}})^{d+2}(Q). \tag{14}$$

Let us check that this condition reduces to the well-known factorization condition in the case of perturbative anomalies. The perturbative part of the anomaly $I \in (I_{\mathbb{Z}}\Omega^{\mathrm{spin}})^{d+2}(BG)$ is given by a characteristic class $I_{d+2}$ constructed from the curvature of the $G$-gauge field and the spacetime metric. Pulling it back to $Q$ sets the differential form $X_{p+1}$ to zero. Then, the condition $p^*(I) = 0$ at the perturbative level means that $I_{d+2}$ vanishes when $X_{p+1}$ is set to zero, i.e. $I_{d+2}$ admits a factorization $I_{d+2} = X_{p+1}Y_{d-p+1}$.

Let us now examine more concretely what this condition (14) means for global anomalies. Suppose we are given a global anomaly $I \in (I_{\mathbb{Z}}\Omega^{\mathrm{spin}})^{d+2}(BG)$. By assumption, this is a torsion element. Let us say that this anomaly corresponds to the eta invariant $\eta(R)$ of a fermion in the representation $R$ and is nonzero when evaluated on a class $[P \to M] \in \Omega_{d+1}^{\mathrm{spin}}(BG)$. Such a class is given by a map $f : M \to BG$, so we denote the class as $[f : M \to BG]$ from now on.

If this map lifts to a map $f' : M \to Q$, such that $f = p \circ f'$, we have a class

$$[f' : M \to Q] \in \Omega_{d+1}^{\mathrm{spin}}(Q), \tag{15}$$

mapping to

$$[f : M \to BG] \in \Omega_{d+1}^{\mathrm{spin}}(BG), \tag{16}$$

by the map

$$p_* : \ \Omega_{d+1}^{\mathrm{spin}}(Q) \to \Omega_{d+1}^{\mathrm{spin}}(BG). \tag{17}$$

Then the pullback $p^*(I)$ evaluated on $[f' : M \to Q]$ is by definition $I = \eta(R)$ evaluated on $[f : M \to BG]$. We thus found that if the eta invariant $\eta(R)$ on $f : M \to BG$ is nonzero and $f : M \to BG$ lifts to $f' : M \to Q$, the anomaly is not canceled. Taking the contrapositive, the condition $p^*(I) = 0$ for a global anomaly $I$ to be canceled by the Green-Schwarz mechanism is that no class $[f : M \to BG] \in \Omega_{d+1}^{\mathrm{spin}}(BG)$ detected by $I$ lifts to a class $[f' : M \to Q] \in \Omega_{d+1}^{\mathrm{spin}}(Q)$.

## 2.4 Generalization

So far we considered the case when the characteristic class $X_{p+1}$ appearing in the modified Bianchi identity (8) was constructed purely from the $G$-gauge field. More generally, there are cases where $X_{p+1}$ involves also terms constructed from the spacetime curvature. Here we briefly outline a generalization to such cases. In practice, we encounter only the case when $p = 3$ and the condition is

$$dH_3 = \frac{p_1}{2} + X'_4, \tag{18}$$

where $p_1$ is the spacetime Pontryagin class and $X'_4$ is a characteristic class of the $G$ gauge field. This is what happens in the case of heterotic string theory, and we concentrate on this subcase in this paper.

When $X'_4 = 0$, the spacetime structure obtained by demanding $dH_3 = p_1/2$ is mathematically known as the string structure, and the bordism group of such manifolds is denoted by $\Omega_d^{\mathrm{string}}(pt)$. With a $G$-bundle satisfying the modified Bianchi identity with $X'_4$ specified by a map $X' : BG \to K(\mathbb{Z}, 4)$, the structure is called the string structure twisted by $X'$, and the bordism group of such manifolds can be denoted by $\Omega_{d;X'}^{\mathrm{string}}(BG)$.[4] Then, the argument as performed in the previous subsection means that a global anomaly $I \in (I_{\mathbb{Z}}\Omega^{\mathrm{spin}})^{d+2}(BG)$ is canceled by the introduction of $H_3$ with the condition (18) if no class $[f : M \to BG] \in \Omega_{d+1}^{\mathrm{spin}}(BG)$ detected by $I$ lifts to a class in $\Omega_{d+1;X'}^{\mathrm{string}}(BG)$, under the natural forgetful map

$$\Omega_{d+1;X'}^{\mathrm{string}}(BG) \to \Omega_{d+1}^{\mathrm{spin}}(BG). \tag{19}$$

In our discussions of concrete cases, we will study two cases, $dH = X_4$ and $dH = p_1/2 + X_4$, for the same characteristic class $X_4 \in H^4(BG; \mathbb{Z})$ in parallel, both in four dimensions and in eight dimensions. Their anomaly cancellation conditions are controlled by the maps (17) and (19), respectively. As there does not seem to be a natural map comparing $\Omega_{d+1}^{\mathrm{spin}}(Q)$ and $\Omega_{d+1;X}^{\mathrm{string}}(BG)$, we need to study them separately. Because of this reason, there does not seem to be a simple relation between whether an anomaly can be canceled by $dH = X_4$ and whether an anomaly can be canceled by $dH = p_1/2 + X_4$.

## 2.5 Comparison to previous works

Before moving on to the analysis of the concrete cases, let us compare the method explained above to the previous works in the literature.

We first note that our condition (14) is not new. It first appeared in the following context. Say that there is an anomaly invertible theory in the bulk $(d+1)$-dimensional space described by a class $I \in H^{d+2}(B\Gamma; \mathbb{Z})$ for a symmetry group $\Gamma$. We consider the fibration $BH \to BG \xrightarrow{p} B\Gamma$ of classifying spaces associated to a group extension $0 \to H \to G \to \Gamma \to 0$. If $p^*(I) = 0$, then the $d$-dimensional boundary $H$-gauge theory can have a $G$-anomaly described by the anomaly theory $I$, and therefore can be used to cancel that. This was first mentioned in [36, Sec.3.3.3] and was further studied in [28, 37]. This was then generalized to the anomaly theories described by bordisms in [20]. The condition (14) for the antisymmetric tensor fields was also used heavily in [31, 32] as a method to see if both the perturbative and global parts of the anomaly of the heterotic string theory compactifications were canceled.

In [23], the following strategy was used instead. Note that the modified Bianchi identity of the antisymmetric tensor field $H_p$ is expressed by the conditions (8) and (9). If we temporarily forget that $X$ is constructed from the $G$-gauge field, and regard it as a general map

---

[4]Twisted string structures and their use in the anomaly cancellation have been explored e.g. in [2,31,32]. We also note related works [3, 19], which explore anomaly cancellations of string theory models using the modern bordism point of view.

$\underline{X} : M \to K(\mathbb{Z}, p + 1)$, we can consider the theory of $H_p$ as a theory defined on spin manifolds, coupled to the background field $\underline{X}$. Using the construction of the antisymmetric tensor field $H_p$ in $d$ dimensions as a boundary theory of a $BF$-type theory in $d + 1$ dimensions, the authors of [23] showed that it can be made to have an arbitrary anomaly $I' \in \widetilde{(I_{\mathbb{Z}}\Omega^{\mathrm{spin}})}^{d+2}(K(\mathbb{Z}, p + 1))$.

Let us now say that $\underline{X}$ is given by composing $f : M \to BG$ and $X : BG \to K(\mathbb{Z}, p + 1)$. Then we can regard this theory of the antisymmetric tensor field to have an anomaly $X^*(I') \in (I_{\mathbb{Z}}\Omega^{\mathrm{spin}})^{d+2}(BG)$, obtained by pulling back $I'$ via $X$. Therefore, an anomaly $I \in (I_{\mathbb{Z}}\Omega^{\mathrm{spin}})^{d+2}(BG)$ of the original theory can be canceled if it is in the image of the pull-back

$$X^* : \widetilde{(I_{\mathbb{Z}}\Omega^{\mathrm{spin}})}^{d+2}(K(\mathbb{Z}, p + 1)) \to \widetilde{(I_{\mathbb{Z}}\Omega^{\mathrm{spin}})}^{d+2}(BG). \tag{20}$$

Note that this is a sufficient condition but not a necessary condition.

Note also that this is compatible with the condition (14) we have derived above, since the fiber sequence (12) ensures that the sequence of pull-backs,

$$\widetilde{(I_{\mathbb{Z}}\Omega^{\mathrm{spin}})}^{d+2}(K(\mathbb{Z}, p + 1)) \xrightarrow{X^*} \widetilde{(I_{\mathbb{Z}}\Omega^{\mathrm{spin}})}^{d+2}(BG) \xrightarrow{p^*} \widetilde{(I_{\mathbb{Z}}\Omega^{\mathrm{spin}})}^{d+2}(Q), \tag{21}$$

satisfies $p^* \circ X^* = 0$. But there can be elements in $(I_{\mathbb{Z}}\Omega^{\mathrm{spin}})^{d+2}(BG)$ which are not in the image of $X^*$, which is still sent to zero by $p^*$, and our condition (14) is more general.

In [9], the authors considered chiral $p$-form fields rather than non-chiral $p$-form fields as we have been considering. Therefore their method and ours cannot be directly compared. Let us simply outline their strategy, which is similar to that of [23]. The authors of [9] identified the possible anomalies of the chiral $p$-form fields coupled to a $(p + 1)$-form background field, following the original analysis of [18]. This gives a subset of $(I_{\mathbb{Z}}\Omega^{\mathrm{spin}})^{d+2}(K(\mathbb{Z}, p + 1))$, unlike in the case of non-chiral $p$-form fields where all anomalies are allowed. This allowed them to cancel the anomalies $I \in (I_{\mathbb{Z}}\Omega^{\mathrm{spin}})^{d+2}(BG)$ which is in the image of the pullback $X^*$ of this subset.

## 3 Four-dimensional Witten anomaly

After the preparations, let us first study if the Witten anomaly of 4d $SU(2)$ and $Sp(n)$ gauge theories can be canceled by the introduction of an antisymmetric tensor field $H_3$.[5] The answer is no, as we can see as follows. Below we only consider the case of $SU(2)$ for simplicity, but the same argument applies to $Sp(n)$, by its embedding $SU(2) \simeq Sp(1) \subset Sp(n)$.

We already discussed in Sec. 2.1 how the Witten anomaly is detected by the generator of $\Omega_5^{\mathrm{spin}}(BSU(2)) = \mathbb{Z}_2$. As explained there, an explicit bordism class realizing this generator is given by a nontrivial $SU(2)$ bundle over $S^5$, obtained by gluing trivial $SU(2)$ bundles on the two hemispheres of $S^5$ via the gauge transformation $g : S^4 \to SU(2)$ on the equator $S^4$ of $S^5$, where $g$ gives the nontrivial element of the homotopy group $\pi_4(SU(2)) = \mathbb{Z}_2$. Let us denote this bordism class as $[f : S^5 \to BSU(2)]$.

### 3.1 With $dH \propto \mathrm{tr}\, F^2$

Let us first consider the cancellation of the anomaly when we introduce an $H$ field such that $dH = c_2$, where $c_2$ is the instanton number density of $SU(2)$. The question then is whether

---

[5]In [14, Sec. 5], it was already argued that the 4d $SU(2)$ Witten anomaly cannot be canceled by topological degrees of freedom alone. This suggests the following argument for the impossibility to cancel the same with the $H$ field. Suppose on the contrary it is possible to do so. We now dualize $H$ to the axion $\phi$. We can then add a random potential $V(\phi)$, leaving only isolated vacua. This would result in the cancellation of Witten's anomaly with only topological degrees of freedom, contradicting [14, Sec. 5]. The authors thank M. Montero for suggesting this argument.

this class $[f : S^5 \to BSU(2)]$ lifts to a class in $\Omega_5^{\mathrm{spin}}(Q)$, where $Q$ is the homotopy fiber of the map $c_2 : BSU(2) \to K(\mathbb{Z}, 4)$ corresponding to the element $c_2 \in H^4(BSU(2), \mathbb{Z})$.

To study it, note that the bordism class $[f : S^5 \to BSU(2)]$ comes from a nontrivial element of $\pi_5(BSU(2)) = \pi_4(SU(2)) = \mathbb{Z}_2$. We now apply the long exact sequence of homotopy groups of the fibration

$$Q \to BSU(2) \to K(\mathbb{Z}, 4), \tag{22}$$

which gives

$$0 = \pi_6(K(\mathbb{Z}, 4)) \to \pi_5(Q) \to \pi_5(BSU(2)) \to \pi_5(K(\mathbb{Z}, 4)) = 0. \tag{23}$$

Therefore[6] the map $f : S^5 \to BSU(2)$ in question lifts to a map $f' : S^5 \to Q$, and the eta invariant still gives a nonzero value on this class. From our argument in Sec. 2.3, this means that the anomaly is not canceled.

Let us make three comments before proceeding. First, we will use spectral sequences to compute and compare the bordism groups to see if the mod-2 anomalies are canceled or not in eight dimensions in the next section. We can equally use the same technique of spectral sequences in the four dimensional case, but it will involve a number of technical issues. We will detail these issues in Appendix A, as they are instructive as a general lesson when spectral sequences are used to answer these questions.

Second, we note that the same nontrivial class of $\Omega_5^{\mathrm{spin}}(BSU(2)) = \mathbb{Z}_2$ has a different representative given by $S^4$ with an $SU(2)$ bundle with instanton number one, times $S^1$ with periodic spin structure.[7] We denote this $G$-bundle by $g : S^4 \times S^1 \to BSU(2)$. Then we have

$$[f : S^5 \to BSU(2)] = [g : S^4 \times S^1 \to BSU(2)] \in \Omega_5^{\mathrm{spin}}(BSU(2)). \tag{24}$$

Now, we showed above that $f$ lifts to a map $f' : S^5 \to Q$, and therefore determines a class in $\Omega_5^{\mathrm{spin}}(Q)$. In contrast, $g : S^4 \times S^1 \to BSU(2)$ does not lift to a map $g' : S^4 \times S^1 \to Q$. To see this, first note that $c_2 \circ g : S^4 \times S^1 \to K(\mathbb{Z}, 4)$ is nontrivial, as we assumed that the instanton number of the $SU(2)$ bundle over $S^4$ is one. Now assume that $g'$ exists. Then $c_2 \circ g = c_2 \circ p \circ g'$, but $c_2 \circ p = 0$ up to homotopy. Therefore $c_2 \circ g$ is zero, which is a contradiction.

This consideration illustrates the point that even when a class $C \in \Omega_d^{\mathrm{spin}}(BG)$ lifts to a class in $\Omega_d^{\mathrm{spin}}(Q)$, it does not mean that all representatives of $C$ of the form $f : M \to BG$ lifts to a $f' : M \to Q$; it suffices to have a single representative $f : M \to BG$ that lifts to $f' : M \to Q$.

Third, the argument we provided for the 4d Witten anomaly can be generalized easily to show that it is impossible to cancel the traditional global anomaly associated to $\pi_d(G)$ in terms of the Green-Schwarz mechanism. We present this generalization in Appendix C.

## 3.2 With $dH \propto \mathrm{tr}\, F^2 - \mathrm{tr}\, R^2$

Before proceeding, let us briefly mention the case when the $H$ field is introduced so that it satisfies $dH = c_2 + p_1/2$, where $p_1$ is the spacetime Pontryagin class. This is what usually appears in the heterotic string construction. In this case, the question is whether the generator of $\Omega_5^{\mathrm{spin}}(BSU(2)) = \mathbb{Z}_2$ lifts to $\Omega_{5;c_2}^{\mathrm{string}}(BSU(2))$. The answer is that it does lift.

Indeed, take the configuration $f : S^5 \to BSU(2)$ above, corresponding to the generator of $\pi_5(BSU(2)) = \mathbb{Z}_2$. By the definition of the twisted string structure, the only

---

[6]Note that the same argument as applied to the fibration $P \to SU(2) = S^3 \to K(\mathbb{Z}, 3)$, where the second map corresponds to the generator of $H^3(S^3, \mathbb{Z})$, shows that $\pi_4(P) = \pi_4(S^3)$. This, combined with the Leray-Serre spectral sequence for $P$, can be used to determine $\pi_4(S^3)$, which was actually the original method used by Serre to determine $\pi_4(S^3)$. We review this classic computation in Appendix B.

[7]In the Adams spectral sequence, this is related to the fact that this class is obtained by multiplying the generator of $\widehat{\Omega_4^{\mathrm{spin}}}(BSU(2)) = \mathbb{Z}$ by the action by $h_1 \in \mathrm{Ext}_{\mathcal{A}(1)}^{1,2}(\mathbb{Z}_2, \mathbb{Z}_2)$.

topological obstruction to find one on a manifold $M$ with a given $SU(2)$ bundle is that $c_2 + p_1/2 \neq 0 \in H^4(M;\mathbb{Z})$. In our case $H^4(S^5;\mathbb{Z}) = 0$, so the configuration above lifts to a twisted string structure. Therefore the anomaly is not canceled in this manner either.

Note that in [2] the group $\Omega_{5;c_2}^{\text{string}}(BSU(2))$ was determined to be $\mathbb{Z}_2$. What we discussed here then means that the natural forgetful map

$$\Omega_{5;c_2}^{\text{string}}(BSU(2)) \to \Omega_5^{\text{spin}}(BSU(2)), \tag{25}$$

is an isomorphism.

# 4 Eight-dimensional mod-2 anomalies

Let us now move on to the mod-2 anomalies of 8d $\mathcal{N}{=}1$ $Sp(n)$ gauge theory. This case is significantly more complicated than in four dimensions. We start by recalling the anomalies of the fermions.

## 4.1 Anomalies of the fermions

**Notations:** Let us begin by setting up our conventions. Recall that $H^*(BSp(n);\mathbb{Z})$ is a polynomial ring $\mathbb{Z}[q_1, q_2, \ldots, q_n]$, where $q_i$ is the $i$-th quaternionic Pontryagin class of degree $4i$. This means that $H^8(BSp(n),\mathbb{Z})$ is generated by $q_1^2$ and $q_2$ for $n \geq 2$ while $q_2$ is absent when $n = 1$. This makes the case of $Sp(1) \simeq SU(2)$ somewhat special. In this paper we will only consider the generic case $n \geq 2$. Our convention is that $q_1 = -c_2$ for $Sp(1) \simeq SU(2)$ and $q_k$ pulls back to the elementary symmetric polynomial of degree $k$ of $q_1$'s when pulled back from $Sp(n)$ to $Sp(1)^n$.

We use the standard decomposition of the bordism group

$$\Omega_9^{\text{spin}}(BSp(n)) \simeq \Omega_9^{\text{spin}}(pt) \oplus \widetilde{\Omega_9^{\text{spin}}}(BSp(n)), \tag{26}$$

into its value at the point and the reduced bordism group. The first summand is well-known to be $\mathbb{Z}_2^2$, generated by $S^1$ with periodic spin structure times the generators of $\Omega_8^{\text{spin}}(pt) = \mathbb{Z}^2$. The second summand was computed in [23] to be $\mathbb{Z}_2^2$. It was also shown there that the following map is surjective:

$$\mathbb{Z}^3 \simeq \widetilde{\Omega_8^{\text{spin}}}(BSp(n)) \xrightarrow{\times(S^1 \text{ with periodic spin structure})} \widetilde{\Omega_9^{\text{spin}}}(BSp(n)) \simeq \mathbb{Z}_2^2. \tag{27}$$

Note that this homomorphism reduces the rank by 1.

**Gravitational part:** For the discussions below it is useful to name explicit generators of these bordism groups, and the indices of the Dirac operators which detect them. The bordism group $\Omega_8^{\text{spin}}(pt) = \mathbb{Z}^2$ are generated by

- $\mathbb{HP}^2$, whose cohomology ring is $H^*(\mathbb{HP}^2;\mathbb{Z}) = \mathbb{Z}[x]/(x^3)$ where $x$ is of degree 4. We note that $\int_{\mathbb{HP}^2} x^2 = 1$. In addition, the Pontryagin classes are given by $p_1 = 2x$ and $p_2 = 7x^2$.

- The Bott manifold $B_8$, which is a manifold with $\int_{B_8} \hat{A} = 1$ and $p_1 = 0$.

They are detected by:

- The standard Dirac operator $D$,

- the Dirac operator $D(TM \ominus \mathbb{R})$ for the spinor bundle tensored with $TM \ominus \mathbb{R}$, i.e. the index of the 8d gravitino.

Here and below, we use the convention that $\ominus$ denotes a formal difference of vector bundles, which corresponds to the reversal of the chirality of the spinors.

**Gauge part:** Next, the generators of $\widetilde{\Omega_8^{\rm spin}}(BSp(n)) = \mathbb{Z}^3$ are

- $\mathbb{HP}^2$ with its canonical $Sp(1)$ bundle, regarded as an $Sp(n)$ bundle. Let it be denoted by $f : \mathbb{HP}^2 \to BSp(n)$. It has $p_1/2 = q_1$ and $\int_{\mathbb{HP}^2} q_1^2 = 1$.

- $\mathbb{HP}^1 \times \mathbb{HP}^1$ with each $\mathbb{HP}^1$ equipped with the canonical quaternionic bundle. This is naturally an $Sp(1) \times Sp(1) \subset Sp(2)$ bundle, which can then be regarded as an $Sp(n)$ bundle. It has $\int_{\mathbb{HP}^1 \times \mathbb{HP}^1} q_1^2 = 2$ and $\int_{\mathbb{HP}^1 \times \mathbb{HP}^1} q_2 = 1$. Let this configuration be denoted by $g : \mathbb{HP}^1 \times \mathbb{HP}^1 \to BSp(n)$.

- $S^8$ with the $Sp(n)$ bundle corresponding to the generator of $\pi_8(BSp(n)) = \mathbb{Z}$. Let it be denoted by $h : S^8 \to BSp(n)$. We have $q_1 = 0 \in H^4(S^8; \mathbb{Z})$, and $\int_{S^8} q_2 = 12$. This factor of 12 will be explained in Sec. 5.1.

We then consider Dirac operators detecting them. We use the notation $D(R)$ for the Dirac operator in the $Sp(n)$ representation $R$. We typically consider $D(\underline{R})$, where we use the abbreviation

$$\underline{R} := R \ominus 1^{\oplus \dim R}, \tag{28}$$

where $\ominus$ means the formal difference of the bundles as stated before. Then $D(\underline{R})$ is for a chiral fermion in $R$ together with $\dim R$ anti-chiral neutral fermions, so that the purely gravitational part of the anomaly cancels. With this proviso, we introduce three Dirac operators as given below:

- $D(\underline{V})$ for the fundamental representation $V$ of $Sp(n)$. We note that this representation is pseudoreal and that the spinor bundle in Euclidean dimension 8 is strictly real. Then the index of $D(\underline{V})$ is automatically even.

- $D(\underline{U})$ for

$$U = \wedge^2 \ominus V^{\oplus 2(n-1)}, \tag{29}$$

where $\wedge^2$ is the antisymmetric square of $V$.

- $D(\underline{Y})$ for

$$Y = S^2 \ominus \wedge^2 \ominus V^{\oplus 4}, \tag{30}$$

where $S^2$ is the symmetric square of $V$.

The pairing of the generators of the bordism group and the Dirac operators can be computed by the index theorem of Atiyah and Singer:

$$\operatorname{ind} D(R) = \int_M \hat{A}_{TM} \operatorname{ch}(R) = \int_M \left( \operatorname{ch}(R)_4 - \frac{p_1}{24} \operatorname{ch}(R)_2 + \frac{7p_1^2 - 4p_2}{5760} \dim R \right). \tag{31}$$

The Dirac indices and the related classes included in the table are obtained through the following procedures from top to bottom. The results are given in Table 1.[8]

---

[8]In more detail, we do the following. For $\operatorname{ind} D$, the index is computed by the third term due to the absence of the gauge part. For $\operatorname{ind} D_{\rm 8d\,grav}$ of the gravitino, we need to replace the gauge strength with the spacetime curvature. So we use the relations of $\operatorname{ch}_2 = p_1$, $\operatorname{ch}_4 = \frac{p_1^2 - 2p_2}{12}$, and $\dim R = 8 - 1 = 7$. For other three cases, $\operatorname{ch}(V)_2 = q_1$, $\operatorname{ch}(V)_4 = \frac{q_1^2 - 2q_2}{12}$. In addition, $\operatorname{ch}(U)_4$ and $\operatorname{ch}(Y)_4$ are the actual values listed in the table, and please note that $\operatorname{ch}_2$s are zero. The other quantities required for the calculation are also zero.

Table 1: The generators of the bordism group $\Omega_8^{\text{spin}}(BSp(n))$, their properties, and the pairing with the Dirac operators.

|  |  | $\mathbb{HP}^2$ | Bott | $[\mathbb{HP}^2, f]$ | $[\mathbb{HP}^1 \times \mathbb{HP}^1, g]$ | $[S^8, h]$ |
|---|---|---|---|---|---|---|
| 0 when multiplied by $S_{\text{periodic}}^1$? |  | No | No | No | No | Yes |
| lifts to $\Omega_8^{\text{spin}}(Q)$? |  | Yes | Yes | No | No | Yes |
| lifts to $\Omega_{8;-q_1}^{\text{string}}(BSp(n))$? |  | No | Yes | Yes | No | Yes |
| $\text{ind}\, D$ | $\frac{7p_1^2 - 4p_2}{5760}$ | 0 | 1 | 0 | 0 | 0 |
| $\text{ind}\, D_{\text{8d grav}}$ | $\frac{289p_1^2 - 988p_2}{5760}$ | $-1$ | 247 | $-1$ | 0 | 0 |
| $\frac{1}{2}\text{ind}\, D(\underline{V})$ | $\frac{q_1}{24}(q_1 - \frac{p_1}{2}) - \frac{q_2}{12}$ | 0 | 0 | 0 | 0 | $-1$ |
| $\text{ind}\, D(\underline{U})$ | $q_2$ | 0 | 0 | 0 | 1 | 12 |
| $\text{ind}\, D(\underline{Y})$ | $q_1^2 - 2q_2$ | 0 | 0 | 1 | 0 | $-24$ |

Note that we listed $\frac{1}{2}\text{ind}\, D(\underline{V})$ in the table, because $\text{ind}\, D(\underline{V})$ is automatically even as explained above. That the $5 \times 5$ integral matrix in the table is invertible guarantees that the five bordism classes and the Dirac indices are dual bases of the non-torsion part of $\Omega_8^{\text{spin}}(BSp(n))$ and $(I_{\mathbb{Z}}\Omega^{\text{spin}})^8(BSp(n))$, respectively.

**Relating 8-dimensional and 9-dimensional bordism groups:** In [23], the bordism group $\Omega_9^{\text{spin}}(BSp(n)) \simeq \mathbb{Z}_2^4$ was also determined. As already mentioned, they are all in the image of the map

$$\mathbb{Z}^5 \simeq \Omega_8^{\text{spin}}(BSp(n)) \xrightarrow{\times (S^1 \text{ with periodic spin structure})} \Omega_9^{\text{spin}}(BSp(n)) \simeq \mathbb{Z}_2^4. \tag{32}$$

In [23], it was found that the kernel of this map is generated by $[S^8, h]$.

To have a better idea why, take an 8-dimensional configuration $f : M_8' \to BSp(n)$ and consider

$$[f : M_8' \to BSp(n)] \times [S_{\text{periodic}}^1] \in \Omega_9^{\text{spin}}(BSp(n)). \tag{33}$$

Then, the exponentiated eta invariant for a Dirac operator in a representation $R$ in such a 9-dimensional configuration satisfies

$$e^{2\pi i \eta(R)} = (-1)^{\text{ind}_{M_8'} D(R)}. \tag{34}$$

Note that $e^{2\pi i \frac{1}{2}\eta(R)}$ is not well-defined even for the case when $R = \underline{V}$. This is because whether the eta invariant in dimension 9 can be improved by a factor $1/2$ depends on whether the index of the Dirac operator $D(R)$ in dimension 10 is always even. Here the spinor bundle in dimension 8 tensored with $\underline{V}$ is pseudoreal and $\text{ind}\, D(\underline{V})$ is always even in dimension 8, but the spinor bundle in dimension 10 tensored with $\underline{V}$ is complex and $\text{ind}\, D(\underline{V})$ is not necessarily even.

Now, a quick glance of Table 1 shows that the only Dirac index with an odd pairing with $[S^8, h]$ is $\frac{1}{2}\text{ind}\, D(\underline{V})$. This means that the Dirac indices appearing in the evaluation of (34) for $[S^8, h] \times [S^1]$ is always even, and therefore no eta invariants detect this bordism class. This is consistent with this class being null.

**Fermion anomalies:** Given a collection of fermions, the anomaly is given by an element in $(I_{\mathbb{Z}}\Omega^{\text{spin}})^{10}(BSp(n))$. As $\Omega_{10}^{\text{spin}}(BSp(n))$ is empty, such an element is characterized by a homomorphism

$$\Omega_9^{\text{spin}}(BSp(n)) \to \mathbb{Z}_2. \tag{35}$$

As argued above, all 9-dimensional configurations in $\Omega_9^{\mathrm{spin}}(BSp(n))$ have representatives given by $S^1_{\mathrm{periodic}}$ times 8-dimensional configurations. Fermion anomalies are given by the eta invariants, which are in turn given by the corresponding eight-dimensional Dirac indices modulo 2 when evaluated on the backgrounds of this form.

Below, we list these Dirac indices modulo 2 for fermions relevant to us.

- First, the anomaly of a neutral fermion is given by the Dirac index $\mathrm{ind}\,D$ modulo 2.

- Second, the anomaly of an 8d gravitino is given by $\mathrm{ind}\,D_{\mathrm{8d\ grav}}$ modulo 2.

- Third, in the $\mathcal{N}{=}1$ $Sp(n)$ theory we have a fermion in the adjoint representation of $Sp(n)$, which is in the symmetric square $S$ of the fundamental representation $V$. We note that $\dim S = n(2n+1)$. Therefore $\mathrm{ind}\,D(S) = \mathrm{ind}\,D(\underline{S}) + n(2n+1)\,\mathrm{ind}\,D$. Recalling the definition of the representations $U$ and $Y$ in (29) and (30), we find that the anomaly of the adjoint fermion in nine dimensions is given by

$$\mathrm{ind}\,D(\mathrm{adj}) \equiv \mathrm{ind}\,D(\underline{U}) + \mathrm{ind}\,D(\underline{Y}) + n\,\mathrm{ind}\,D\,, \tag{36}$$

modulo 2.

## 4.2 Anomaly cancellation

We can now finally discuss whether the anomaly of the 8d system is canceled by the introduction of the three-form field $H_3$. We discuss two cases of the modified Bianchi identities, one given by $dH_3 = q_1$ and the other given by $dH_3 = p_1/2 - q_1$.

**The case $dH_3 = q_1$:** The question is whether the generators of $\widetilde{\Omega_9^{\mathrm{spin}}}(BSp(n))$ can be lifted to $\widetilde{\Omega_9^{\mathrm{spin}}}(Q)$ or not. The latter bordism group will be determined in Sec. 5 to be $\mathbb{Z}_2$, and the map

$$\mathbb{Z}_2 \simeq \widetilde{\Omega_9^{\mathrm{spin}}}(Q) \to \widetilde{\Omega_9^{\mathrm{spin}}}(BSp(n)) \simeq \mathbb{Z}_2^2\,, \tag{37}$$

will be shown to be the zero map. Therefore, there is no way to lift the elements of $\widetilde{\Omega_9^{\mathrm{spin}}}(BSp(n))$ to $\widetilde{\Omega_9^{\mathrm{spin}}}(Q)$. Applying our central result on cancellation in Sec. 2.3, we see that all global purely gauge or mixed gauge-gravity anomalies can be canceled.

Let us compare the finding of [23] again. We already gave a comparison of methods in Sec. 2.5 in a general setting; here we would like to comment on this specific case. In their approach, they considered the map $BSp(n) \to K(\mathbb{Z}, 4)$ classifying the class $q_1 \in H^4(BSp(n); \mathbb{Z})$. Under this map, they found

$$\mathbb{Z}_2^2 \simeq \widetilde{\Omega_9^{\mathrm{spin}}}(BSp(n)) \to \widetilde{\Omega_9^{\mathrm{spin}}}(K(\mathbb{Z}, 4)) \simeq \mathbb{Z}_2\,, \tag{38}$$

where the generator of the image is given by the class $[\mathbb{HP}^2, f] \times [S^1_{\mathrm{periodic}}]$. They also constructed a theory of a 3-form field $H_3$ coupled to the background 4-form field classified by $K(\mathbb{Z}, 4)$, which carries an anomaly detecting the class just discussed above. This allowed them to cancel a $\mathbb{Z}_2$ in the anomaly, but an uncanceled $\mathbb{Z}_2$ remained. Our improvement upon their result was to investigate the map (37) instead of (38), which allowed us to conclude that the entire anomalies are canceled.

**The case $dH_3 = p_1/2 - q_1$:** The question in this case is whether the generators of $\Omega_9^{\text{spin}}(BSp(n))$ lift to the twisted string bordism group $\Omega_{9;-q_1}^{\text{string}}(BSp(n))$ or not, under the natural forgetful map

$$f : \Omega_{9;-q_1}^{\text{string}}(BSp(n)) \to \Omega_9^{\text{spin}}(BSp(n)). \tag{39}$$

The twisted string bordism group $\Omega_{9;-q_1}^{\text{string}}(BSp(n))$ was determined in [2][9] to be $\mathbb{Z}_2^2$. The map

$$\mathbb{Z}^3 \simeq \Omega_{8;-q_1}^{\text{string}}(BSp(n)) \xrightarrow{\times (S^1 \text{ with periodic spin structure})} \Omega_{9;-q_1}^{\text{string}}(BSp(n)) \simeq \mathbb{Z}_2^2 \tag{40}$$

is still surjective, as was in (32). The generators in eight dimensions are the Bott manifold, $[\mathbb{HP}^2, f]$ and $[S^8, h]$, and the kernel is $[S^8, h]$. The generators in nine dimensions are then

$$[\text{Bott}] \times [S_{\text{periodic}}^1], \quad \text{and} \quad [\mathbb{HP}^2, f] \times [S_{\text{periodic}}^1]. \tag{41}$$

A fermion anomaly before the introduction of the three-form field $H_3$ was characterized by a homomorphism (35):

$$I : \Omega_9^{\text{spin}}(BSp(n)) \to \mathbb{Z}_2. \tag{42}$$

With the three-form field $H_3$, it is given by the homomorphism

$$I' : \Omega_{9;-q_1}^{\text{string}}(BSp(n)) \xrightarrow{f} \Omega_9^{\text{spin}}(BSp(n)) \xrightarrow{I} \mathbb{Z}_2, \tag{43}$$

obtained by composing (39) and (42). The homomorphism (42) was given by the eight-dimensional Dirac index modulo 2, as we discussed around (35). Therefore, to compute $I'$ simply means to evaluate these Dirac indices modulo 2 on two generators (41).

From Table 1, it is easy to see that these indices modulo 2 are given by

|  | $[\text{Bott}] \times [S_{\text{periodic}}^1]$ | $[\mathbb{HP}^2, f] \times [S_{\text{periodic}}^1]$ |
|---|---|---|
| $\text{ind}\, D$ | 1 | 0 |
| $\text{ind}\, D_{\text{8d grav}}$ | 1 | 1 |
| $\text{ind}\, D(\underline{Y})$ | 0 | 1 |
| $\text{ind}\, D(\underline{U})$ | 0 | 0 |
| $\text{ind}\, D(\underline{V})$ | 0 | 0 |

$$\tag{44}$$

where the Dirac indices and the entries are considered modulo 2.

From (36), we see that the anomaly of an adjoint fermion of $Sp(n)$ in nine dimensions after the introduction of the three-form field $H_3$ is given modulo 2 by

$$\text{ind}\, D(\text{adj}) \equiv \text{ind}\, D(\underline{Y}) + n\, \text{ind}\, D, \tag{45}$$

which is nonzero when evaluated on the generators (41).

However, the actual 8d $\mathcal{N}=1$ $Sp(n)$ theory one obtains in the heterotic compactification on $T^2$ without vector structure introduced in [35] is a very specific combination of $\mathcal{N}=1$ supergravity theory with $Sp(8)$ gauge group, which contains as its massless fermion spectrum

- three dilatino,[10]

- one gravitino, and

- one adjoint fermion of $Sp(8)$.

---

[9]Note that there is unfortunately an error in its v2 concerning this bordism group, which is corrected in its v3 on the arXiv. The authors thank the authors of [2] for discussions concerning these points.

[10]One is from 10d $\mathcal{N} = 1$ supergravity-multiplet, and two are due to dimensional reduction to 8d.

The total anomaly is then given modulo 2 by

$$\operatorname{ind} D + \operatorname{ind} D_{\text{8d grav}} + \operatorname{ind} D(\text{adj}) \equiv 0, \tag{46}$$

when evaluated on the generators (41), as can be readily checked using the table (44). This means that the global anomaly of the actual 8d $\mathcal{N}=1$ $Sp(8)$ theory obtained in the heterotic compactification is canceled by the introduction of $H_3$. Note that the rank of the $Sp$ gauge group did matter in this computation.

# 5   Determination of the map $\Omega_9^{\text{spin}}(Q) \to \Omega_9^{\text{spin}}(BSp(n))$

In this technical section, we perform various algebraic topological computations to determine the spin bordism group of the space $Q$ introduced previously, and the crucial homomorphism $\Omega_9^{\text{spin}}(Q) \to \Omega_9^{\text{spin}}(BSp(n))$. As the case $n = 1$ is somewhat special, we restrict our attention to $n \geq 2$.[11]

## 5.1   A differential geometric lemma

Before we move on to algebraic topology, we need one input from differential geometrical considerations. Take a spin 8-manifold $M$ with an $Sp(n)$ bundle. We already discussed in Sec. 4.1 the Dirac operator $D(\underline{V})$ for the fundamental representation $V$ of $Sp(n)$, where the purely gravitational part is already subtracted. We have

$$\operatorname{ind} D(\underline{V}) = \int_M \left( \frac{q_1}{12} \left( q_1 - \frac{p_1}{2} \right) - \frac{q_2}{6} \right), \tag{47}$$

which is automatically even because $V$ is pseudoreal and the eight-dimensional spinor is strictly real. From this we learn that $\int_M q_2$ is divisible by 12 if $q_1$ or $q_1 - p_1/2$ is zero.

This information will be used later in this section. In particular, a factor of three will be observed in Sec. 5.2 and a factor of four will be found in Sec. 5.4.

This information can also be used to show why $\int_{S^8} q_2 = 12$ in the configuration $[S^8, h]$ introduced in Sec. 4.1, corresponding to the generator of $\pi_8(BSp) \simeq \mathbb{Z}$. We recall that the index theorem of Atiyah and Singer equates the analytic index and the topological index. Here the topological index takes values in $KSp^0(S^8) \simeq \mathbb{Z}$, and it is one almost by definition. The analytic index is the index of the Dirac operator divided by two in the case of pseudoreal representations, which is therefore $D(\underline{V})/2$. As $q_1 = p_1/2 = 0$ on $S^8$, we conclude that $\int_{S^8} q_2 = 12$.

## 5.2   Cohomology ring of $Q$

Let us determine the cohomology ring of $Q$ to degree 10.

We first need the data of $H^*(K(\mathbb{Z},3);\mathbb{Z})$, which can be found in [4]. As a graded Abelian group, it is given by

$$
\begin{array}{c|ccccccccccc}
d & 0 & 1 & 2 & 3 & 4 & 5 & 6 & 7 & 8 & 9 & 10 \\
\hline
H^d(K(\mathbb{Z},3);\mathbb{Z}) & \mathbb{Z} & 0 & 0 & \mathbb{Z} & 0 & 0 & \mathbb{Z}_2 & 0 & \mathbb{Z}_3 & \mathbb{Z}_2 & \mathbb{Z}_2
\end{array}. \tag{48}
$$

Let us name the generator of degree $d$ as $u_d$. Then we have $u_3 u_6 = u_9$.

Next we need the data of $H^*(BSp(n);\mathbb{Z})$. As already recalled, this is $\mathbb{Z}[q_1, q_2]$ up to degree 10, where the degree of $q_i$ is $4i$.

---

[11] $\Omega_{d \leq 7}^{\text{spin}}(Q)$ for $n = 1$ was determined in [22] in the context of the analysis of global anomalies in six dimensions.

We now use the Leray-Serre spectral sequence for the fibration

$$K(\mathbb{Z}, 3) \to Q \to BSp(n), \tag{49}$$

whose $E_2$-page has the form

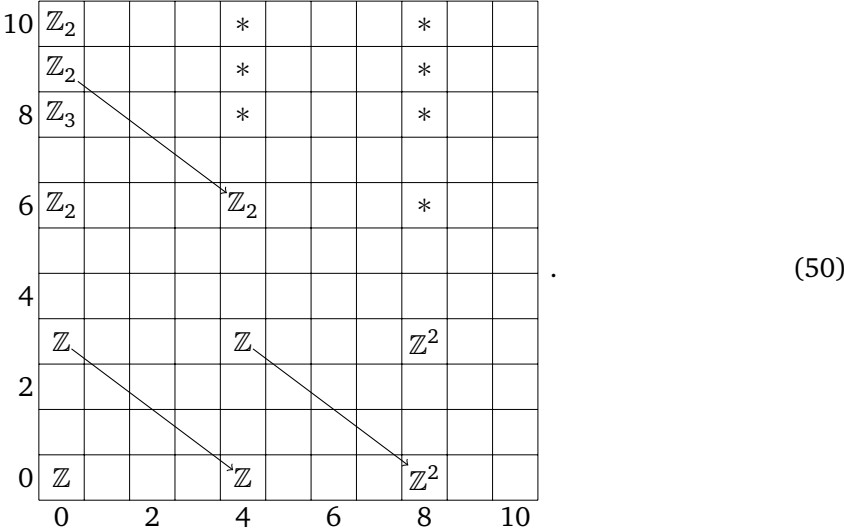

Here we use the convention that an empty square has a zero group in it, and a square with $*$ has an Abelian group which does not concern us.[12]

Clearly $E_2 = E_3 = E_4$ and the first nonzero differential is $d_4$. As $H^4(Q; \mathbb{Z}) = 0$ by design, the differential from $E_2^{0,3}$ to $E_2^{4,0}$ is the identity. This determines other nonzero differentials in this range, using the fact that differentials $d_n$ in the Serre spectral sequence are derivations of $E_n^{p,q}$ as a graded ring. These differentials were already indicated above.

We note that $E_4^{8,0}$ is generated by $q_1^2$ and $q_2$, and the differential hits $q_1^2$. We then find that the $E_5$ page is

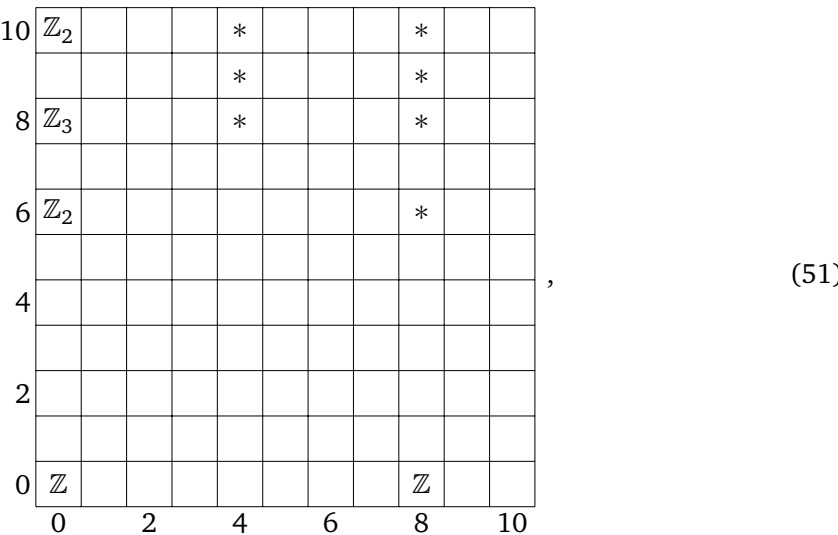

where the generator of $E_5^{8,0}$ is $q_2$. From this we know $H^*(Q; \mathbb{Z})$ as an Abelian group up to degree 10, once we settle the extension problem at degree 8:

$$0 \to \mathbb{Z} \to H^8(Q; \mathbb{Z}) \to \mathbb{Z}_3 \to 0. \tag{52}$$

---

[12]We note that the analogous computation of the Leray-Serre spectral sequence for the fibration $K(\mathbb{Z}, 2) \to P \to Sp(n)$ can be used to determine $\pi_4(Sp(n))$, including $\pi_4(S^3)$. We review this classic computation in Appendix B.

The solution is either a direct sum $\mathbb{Z} \oplus \mathbb{Z}_3$ or $\mathbb{Z}$ with the map $\mathbb{Z} \to H^8(Q;\mathbb{Z})$ given by a multiplication by 3. The correct answer is the latter.

To see this, we use the relation

$$\mathcal{P}^1 r_3(q_1) = -r_3(q_1^2 + q_2), \tag{53}$$

where $\mathcal{P}^1$ is the mod-3 Steenrod power and $r_3$ is the reduction mod 3.[13] Now, $q_1$ is zero when pulled back to $Q$. The relation (53) then implies that the mod-3 reduction of $q_2$ is also zero when pulled back to $Q$. In the two possibilities of the solution to the extension problem, only the latter is consistent with this fact.

We summarize what we found below:

$$
\begin{array}{c|ccccccccccc}
d & 0 & 1 & 2 & 3 & 4 & 5 & 6 & 7 & 8 & 9 & 10 \\
\hline
H^d(Q;\mathbb{Z}) & \mathbb{Z} & 0 & 0 & 0 & 0 & 0 & \mathbb{Z}_2 & 0 & \mathbb{Z} & 0 & \mathbb{Z}_2
\end{array}, \tag{54}
$$

where the generator of $H^8(Q;\mathbb{Z})$ can be denoted by $q_2/3$. The mod-2 cohomology is then

$$
\begin{array}{c|ccccccccccc}
d & 0 & 1 & 2 & 3 & 4 & 5 & 6 & 7 & 8 & 9 & 10 \\
\hline
H^d(Q;\mathbb{Z}_2) & \mathbb{Z}_2 & 0 & 0 & 0 & 0 & \mathbb{Z}_2 & \mathbb{Z}_2 & 0 & \mathbb{Z}_2 & \mathbb{Z}_2 & \mathbb{Z}_2
\end{array}. \tag{55}
$$

Let $u_d$ for $d = 0, 5, 6, 8, 9, 10$ be the generators of $H^d(Q;\mathbb{Z}_2) = \mathbb{Z}_2$.

To use the Adams spectral sequence, we need to determine the action of $\mathrm{Sq}^1$ and $\mathrm{Sq}^2$, where $\mathrm{Sq}^1$ is identified with the Bockstein homomorphism. The action of $\mathrm{Sq}^1$ is determined if the class lifts to an element of $H^*(Q;\mathbb{Z})$, so the nonzero action is $\mathrm{Sq}^1 u_5 = u_6$ and $\mathrm{Sq}^1 u_9 = u_{10}$. For $\mathrm{Sq}^2$, the only possibly nonzero ones in this range are $\mathrm{Sq}^2 u_6$ and $\mathrm{Sq}^2 u_8$. The latter is zero, since $u_8$ is pulled back from $BSp(n)$, for which the action of $\mathrm{Sq}^2$ is zero since $H^{10}(BSp(n);\mathbb{Z}_2) = 0$. The most subtle one is $\mathrm{Sq}^2 u_6$. This turns out to be nonzero and is equal to $u_8$. To see this we use the Atiyah-Hirzebruch spectral sequence and combine it with the differential geometric lemma we discussed above.

## 5.3 Atiyah-Hirzebruch spectral sequence for $\Omega_*^{\mathrm{spin}}(Q)$

We now examine the Atiyah-Hirzebruch spectral sequence for the spin bordism group of $Q$. The $E^2$ page is given as follows:

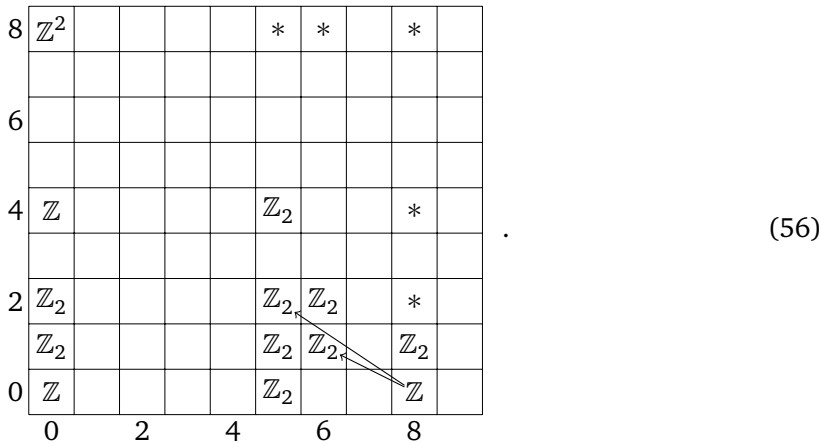

$$. \tag{56}$$

---

[13] This can be found by considering the maximal torus $T^n$ of $Sp(n)$. Let us consider the natural inclusion $i : BT^n \to BSp(n)$. At the cohomology level, this induces an injection $i^* : H^*(BSp(n);\mathbb{Z}_3) \to H^*(BT^n;\mathbb{Z}_3)$. Note that $H^*(BT^n;\mathbb{Z}_3) = \mathbb{Z}_3[t_1,\ldots,t_n]$ and $H^*(BSp(n);\mathbb{Z}_3) = \mathbb{Z}_3[q_1,\ldots,q_n]$ where these generators satisfy the relations $i^*(q_i) = \sigma_i(t_1^2,\ldots,t_n^2)$ where $\sigma_i$ is the $i$-th symmetric polynomial. Within $H^*(BT^n;\mathbb{Z}_3)$, the expression $\mathcal{P}^1(\sum_i t_i^2) = 2\sum_i t_i \mathcal{P}^1(t_i) = 2\sum_i t_i^4 = 2(\sum_i t_i^2)^2 - 4\sum_{i<j} t_i^2 t_j^2$ follow, where in the first and second equations we use the Cartan formula and the definition of the mod-3 Steenrod power respectively. Thus, we obtain $\mathcal{P}^1(\sigma_1) = 2\sigma_1^2 - 4\sigma_2$ by omitting the argument of $\sigma_i$. Since $i^*$ is injective and the action of $\mathcal{P}^1$ is expressed solely in terms of $\sigma_i$, the above expression can be represented using the generators of $H^*(BSp(n);\mathbb{Z})$. As a result, we get $\mathcal{P}^1 r_3(q_1) = r_3(2q_1^2 - 4q_2) = -r_3(q_1^2 + q_2)$ that we wished to obtain.

The generator of $E^2_{8,0}$ is dual to $q_2/3$, i.e. its pairing with $q_2$ is 3. From the discussion in Sec. 5.1, the pairing of $q_2$ with the elements of $E^2_{8,0}$ which survive to $E^\infty$ is divisible by 12. The only manner in which this is achieved in the spectral sequence is that the differentials indicated in (56) above, $d^2 : E^2_{8,0} \to E^2_{6,1}$ and $d^3 : E^3_{8,0} \to E^3_{5,2}$ are both nontrivial. As $d^2 : E^2_{p,0} \to E^2_{p-2,1}$ is the dual of the Steenrod square $\mathrm{Sq}^2$ combined with the mod-2 reduction [29,38], this means that $\mathrm{Sq}^2 u_6$ is nonzero and equal to $u_8$. We can continue the analysis of the Atiyah-Hirzebruch spectral sequence further, but it is now easier to turn to the Adams spectral sequence using this information.

## 5.4 Adams spectral sequence for $\Omega^{\mathrm{spin}}_*(Q)$

In this section, we compute the bordism group $\Omega^{\mathrm{spin}}_*(Q)$ using the Adams spectral sequence. While the Adams spectral sequence is complementary to the Atiyah-Hirzebruch spectral sequence, it tends to be more powerful in higher degree. For the computation of the Adams spectral sequences in the context of anomalies, we refer the readers to [1].

The $E_2$ page of the Adams spectral sequence specialized for the computation of reduced spin bordism groups is given as follows:

$$E^{s,t}_2 = \mathrm{Ext}^{s,t}_{\mathcal{A}}(\widetilde{H^*}(MSpin \wedge X; \mathbb{Z}_2), \mathbb{Z}_2) \Rightarrow \widetilde{\Omega^{\mathrm{spin}}_{t-s}}(X)^\wedge_2, \tag{57}$$

where $\mathcal{A}$ is the mod 2 Steenrod algebra and $MSpin$ is the Thom spectrum of the universal bundle over $BSpin$. Note that this sequence converges to the 2-completion of the bordism groups.

According to [23], the $E_2$ page simplifies further:

$$E^{s,t}_2 = \mathrm{Ext}^{s,t}_{\mathcal{A}(1)}((\mathbb{Z}_2 \oplus \Sigma^8 \mathbb{Z}_2 \oplus M_{\geq 10}) \otimes_{\mathbb{Z}_2} \widetilde{H^*}(X; \mathbb{Z}_2), \mathbb{Z}_2), \tag{58}$$

where $\mathcal{A}(1)$ is the subalgebra of $\mathcal{A}$ generated by $Sq^1$ and $Sq^2$, and $M_{\geq 10}$ is concentrated in degrees 10 and above. Fortunately, within the range needed for our computations, the $E_2$ page can be simplified even more. From equation (55), the reduced cohomology group of $Q$ is trivial up to degree 4. Therefore, we have

$$(\mathbb{Z}_2 \oplus \Sigma^8 \mathbb{Z}_2 \oplus M_{\geq 10}) \otimes_{\mathbb{Z}_2} \widetilde{H^*}(X; \mathbb{Z}_2) \simeq \widetilde{H^*}(X; \mathbb{Z}_2) \oplus M'_{\geq 13}. \tag{59}$$

Consequently, within the range we are focusing on, we obtain:

$$E^{s,t}_2 = \mathrm{Ext}^{s,t}_{\mathcal{A}(1)}(\widetilde{H^*}(X; \mathbb{Z}_2), \mathbb{Z}_2). \tag{60}$$

In the previous two sections, we determined the action of $Sq^1$ and $Sq^2$, on the generators of $\widetilde{H^*}(Q; \mathbb{Z}_2)$ within the range of degree 10 or lower. We separate

$$0 \to M_{\geq 11} \to \widetilde{H^*}(Q; \mathbb{Z}_2) \to M_{\leq 10} \to 0, \tag{61}$$

where $M_{\geq 11}$ is the $\mathcal{A}(1)$ submodule with degree $\geq 11$ and $M_{\leq 10}$ is the quotient concentrated in degree $\leq 10$.

We have determined the $\mathcal{A}(1)$ module structure of $M_{\leq 10}$:

$$\begin{array}{c}
\bullet\, u_{10} \\
\bullet\, u_9 \\
\\
\bullet\, u_8 \\
\\
\bullet\, u_6 \\
\bullet\, u_5
\end{array} \quad , \tag{62}$$

where the straight lines and curved lines represent the actions of $Sq^1$ and $Sq^2$ respectively. Note that this is a direct sum of two submodules, shown in black and magenta, respectively.

The corresponding Adams chart for $M_{\leq 10}$ is given by

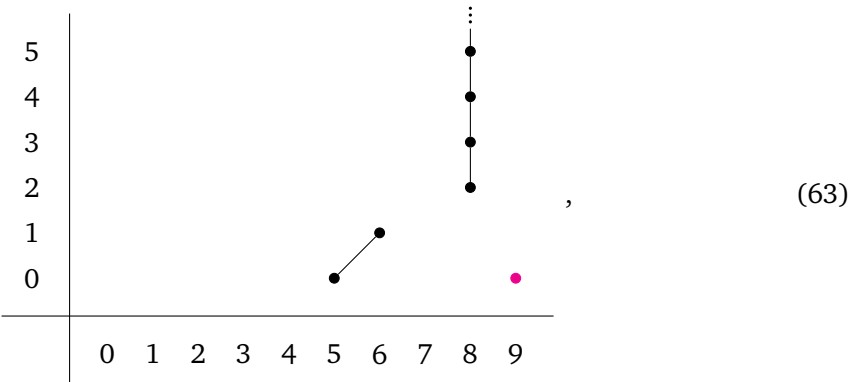

$$\text{,} \tag{63}$$

where the vertical and the horizontal axes are $s$ and $t-s$ respectively, and the vertical and diagonal lines represent the action by $h_0 \in \text{Ext}^{1,1}_{\mathcal{A}(1)}(\mathbb{Z}_2, \mathbb{Z}_2)$ and $h_1 \in \text{Ext}^{1,2}_{\mathcal{A}(1)}(\mathbb{Z}_2, \mathbb{Z}_2)$ respectively. We also distinguished the contributions from two submodules in $M_{\leq 10}$ using the corresponding colors.

Note that we need to investigate whether the module structure of degree 11 or higher affects the computations of the bordism group which we are interested in. To study it, let $M$ be the following $\mathcal{A}(1)$ module

$$\text{,} \tag{64}$$

where the bottom element is at degree 0. We note $\widetilde{H^*}(\mathbb{R}P^2; \mathbb{Z}_2) \simeq \Sigma M$.

Let $X$ be the degree $\geq 9$ part of $\widetilde{H^*}(Q; \mathbb{Z}_2)$ as the $\mathcal{A}(1)$ module. It sits in the short exact sequence of $\mathcal{A}(1)$ modules of the following form

$$0 \to M_{\geq 11} \to X \to \Sigma^9 M \to 0. \tag{65}$$

This short exact sequence induces the following long exact sequences on the $E_2$ page of the Adams spectral sequence [1]:

$$\cdots \longrightarrow \text{Ext}^{s,t}_{\mathcal{A}(1)}(\Sigma^9 M, \mathbb{Z}_2) \longrightarrow \text{Ext}^{s,t}_{\mathcal{A}(1)}(X, \mathbb{Z}_2) \longrightarrow \text{Ext}^{s,t}_{\mathcal{A}(1)}(M_{\geq 11}, \mathbb{Z}_2)$$

$$\xrightarrow{\quad\delta\quad}$$

$$\text{Ext}^{s+1,t}_{\mathcal{A}(1)}(\Sigma^9 M, \mathbb{Z}_2) \longrightarrow \text{Ext}^{s+1,t}_{\mathcal{A}(1)}(X, \mathbb{Z}_2) \longrightarrow \cdots \tag{66}$$

By setting $s = i$ and $t = i + 9$ where $i \geq 0$, we can consider the following diagrams that give isomorphisms:

$$0 \to \text{Ext}^{i,i+9}_{\mathcal{A}(1)}(\Sigma^9 M, \mathbb{Z}_2) \to \text{Ext}^{i,i+9}_{\mathcal{A}(1)}(X, \mathbb{Z}_2) \to 0, \tag{67}$$

where we use the fact that $\text{Ext}^{i-1,i+9}_{\mathcal{A}(1)}(M_{\geq 11}, \mathbb{Z}_2) = \text{Ext}^{i,i+9}_{\mathcal{A}(1)}(M_{\geq 11}, \mathbb{Z}_2) = 0$ because the degrees of the corresponding $t-s$ is less than 11 from the definition of $M_{\geq 11}$. Thus, within the scope of $t-s = 9$, the $E_2$ page of $X$ coincides with that of $\Sigma^9 M$. Therefore, the Adams chart given above, in degrees 9 and below, is independent of the additional structure of $X$.

There is no non-trivial differential in the $E_2$ page, because differentials must commute with the action of $h_0$ and $h_1$. So the 2-completion of the reduced spin bordism groups of $Q$ is given

by

$$
\begin{array}{c|ccccccccccc}
d & 0 & 1 & 2 & 3 & 4 & 5 & 6 & 7 & 8 & 9 & \cdots \\
\hline
\widetilde{\Omega}^{\mathrm{spin}}_d(Q) & 0 & 0 & 0 & 0 & 0 & \mathbb{Z}_2 & \mathbb{Z}_2 & 0 & \mathbb{Z}_2^{\wedge} & \mathbb{Z}_2 & \cdots
\end{array}. \tag{68}
$$

From the Atiyah-Hirzebruch spectral sequence (56), we clearly cannot have any nontrivial structure at the prime $p > 2$, and we conclude that the spin bordism groups of $Q$ before the 2-completion is given by (68) after a replacement of $\mathbb{Z}_2^{\wedge}$ by $\mathbb{Z}$. Note that this $\mathbb{Z}$, or $\mathbb{Z}_2^{\wedge}$, came from the Adams filtration starting $s = 2$ in the Adams spectral sequence, see (63). This means that there is a factor of four in the Hurewicz homomorphism $\widetilde{\Omega}^{\mathrm{spin}}_9(Q) \to H_9(Q;\mathbb{Z})$. Together with the factor of three in $H_9(Q;\mathbb{Z}) \to H_9(BSp(n);\mathbb{Z})$ we already saw at the end of Sec. 5.2, we reproduce the factor 12 we started our discussion in Sec. 5.1.

## 5.5 Behavior under the map $Q \to BSp(n)$

Here, we examine the behavior of the map

$$
p_* : \mathbb{Z}_2 \simeq \widetilde{\Omega}^{\mathrm{spin}}_9(Q) \to \widetilde{\Omega}^{\mathrm{spin}}_9(BSp(n)) \simeq \mathbb{Z}_2^2, \tag{69}
$$

introduced in (37). As we pointed out earlier, this will be the zero map. This is also the result we need to use in Sec. 4.2 to show that the anomaly cancels.

First, we need to recognize that the elements $u_9$ and $u_{10}$ of the cohomology of $Q$, the dots given in magenta in (62), originates from the cohomology of $K(\mathbb{Z},3)$. This can be understood from the fact that in the Leray-Serre spectral sequence of (50) and (51) the $E_2^{0,10}$ component remains up to the $E_\infty$ page. In fact, it also appears in the degrees 9 and 10 in (55).

Next, let us consider the $E_2$ page of the Adams spectral sequence. As mentioned above, in the fibration

$$
K(\mathbb{Z},3) \xrightarrow{\iota} Q \xrightarrow{p} BSp(n), \tag{70}
$$

the elements $u_9, u_{10}$ of $Q$ are pulled back from $K(\mathbb{Z},3)$. This fact partially induces the following exact sequence on the $E_2$ page:

$$
\mathrm{Ext}^{0,9}_{\mathcal{A}(1)}(K(\mathbb{Z},3),\mathbb{Z}_2) \xrightarrow{\simeq} \mathrm{Ext}^{0,9}_{\mathcal{A}(1)}(Q,\mathbb{Z}_2) \longrightarrow \mathrm{Ext}^{0,9}_{\mathcal{A}(1)}(BSp(n),\mathbb{Z}_2). \tag{71}
$$

This in particular means that the bordism class generating $\widetilde{\Omega}^{\mathrm{spin}}_9(Q) \simeq \mathbb{Z}_2$ came from $\widetilde{\Omega}^{\mathrm{spin}}_9(K(\mathbb{Z},3))$ via $\iota_*$, and therefore has a representative of the form $[M_9, f]$ where $f$ is a map $f : M_9 \to K(\mathbb{Z},3)$. Because we have the fibration (70), this map $f$ is null when sent to $BSp(n)$ via $p_*$, and therefore $[M_9, f]$ becomes a zero class in $\widetilde{\Omega}^{\mathrm{spin}}_9(BSp(n))$. This means that the map $p_*$ in (69) is the zero map.

Note that we cannot simply use the exactness of (71), or the fact that $\mathrm{Ext}^{0,9}_{\mathcal{A}(1)}(BSp(n),\mathbb{Z}_2)$ is zero, to conclude that the map $p_*$ is the zero map. This is due to the extension problem determining the bordism group from the $E_\infty$ page. Indeed, we have the following commuting diagram involving two short exact sequences:

$$
\begin{array}{c|ccccccccc}
& 0 \to & E^{1,10} & \to & \Omega^{\mathrm{spin}}_9 & \to & E^{0,9} & \to 0 \\
\hline
Q & 0 \to & 0 & \to & \mathbb{Z}_2 & \to & \mathbb{Z}_2 & \to 0 \\
& & & & \downarrow & & \downarrow p_* & & \downarrow \\
BSp(n) & 0 \to & \mathbb{Z}_2^2 & \to & \mathbb{Z}_2^2 & \to & 0 & \to 0
\end{array}, \tag{72}
$$

and any homomorphism $\mathbb{Z}_2 \to \mathbb{Z}_2^2$ is compatible with the commutativity. In this case, $p_*$ turns out to be zero, but when we apply a similar spectral sequence technique to study $\Omega^{\mathrm{spin}}_5(Q) \to \Omega^{\mathrm{spin}}_5(BSp(n))$, a homomorphism in the same position turns out to be a nonzero map, see (A.12).

# 6 Conclusions and discussions

In this paper, we mainly studied the question of when a mod-2 global anomaly for a fermion can be canceled by the introduction of a three-form field $H_3$ with a modified Bianchi identity $dH_3 \propto \operatorname{tr} F^2$. This question can be considered as the global anomaly version of the Green-Schwarz cancellation mechanism, which are usually only presented at the perturbative level in the literature. In our method, we first represent the configuration space of the $H$ field together with the $G$ gauge field as a fibration $Q$ of the Eilenberg-Mac Lane space $K(\mathbb{Z}, 3)$ over $BG$ obtained by taking the homotopy fiber of the map $BG \to K(\mathbb{Z}, 4)$ characterizing the modified Bianchi identity. Then the anomaly can be canceled if and only if the anomaly class in $\Omega_{d+1}^{\text{spin}}(BG)$ does not lift to $\Omega_{d+1}^{\text{spin}}(Q)$. Then, the remaining issue was a careful study of this bordism group $\Omega_{d+1}^{\text{spin}}(Q)$ in terms of Atiyah-Hirzebruch or Adams spectral sequences.

We then illustrated our general arguments by two concrete examples, by studying whether the mod-2 $Sp(n)$ anomalies in four dimensions and in eight dimensions can be canceled or not. We found that the 4d anomaly cannot be canceled, whereas the 8d one can be canceled. We also discussed, albeit briefly, what happens if we consider the modified Bianchi identity of the form $dH_3 = a \operatorname{tr} F^2 + b \operatorname{tr} R^2/2$, using twisted string structure.

Before closing, let us discuss possible extensions and applications of the techniques developed in this paper. First, in this paper, we only discussed cancelling mod-2 anomalies. Generalizing this analysis to the cancellation of more general $\mathbb{Z}_k$ anomalies can be done immediately. In fact, the general bordism framework to treat anomalies do not single out $\mathbb{Z}_2$ out of other discrete global anomalies. If we use the Anderson dual of bordism group instead of the bordism group itself as expounded in [12], it does not even distinguish the torsion (or equivalently global anomaly) part and the non-torsion (or equivalently perturbative anomaly) part, either. Therefore, our method and our criterion are general enough and applicable to any discussion of both global and perturbative anomaly cancellation by a continuous $p$-form field.

In the presence of gravity, the modified Bianchi identity can still be of the form $dH \propto \operatorname{tr} F^2$ (as in Type IIB supergravity in the presence of both D9 and anti D9-branes) but can also be of the form $dH \propto a \operatorname{tr} F^2 + b \operatorname{tr} R^2$ (as in Type I or heterotic supergravity). In the latter case, in practical applications, only a very specific coefficient $b$ has ever been encountered, which corresponds to trivializing $p_1(R)/2$, i.e. one half of the first Pontryagin class of the spacetime, when the gauge field is turned off. In this case, the configuration space to be considered is not simply a $K(\mathbb{Z}, 3)$ fibration over $BG$ but is the classifying space of the totality of the spacetime bordism class, the gauge field, and the $H$ field with the modified Bianchi identity. Mathematically, this boils down to the computation of the twisted string bordism groups, which have been performed in many of the cases relevant to various string theory constructions in the recent literature, e.g. [2, 5, 10, 11]. To the knowledge of the authors, no attempts have been made to generalize these computations to the case when the spacetime contribution to the modified Bianchi identity is *not* equal to that of a twisted string structure. Without a concrete system of this type in some string theory consideration, there would not be much incentive to study such cases, although an investigation using a similar technique of algebraic topology would surely be possible.

One important caveat to be noted on the discussions so far here is that we always assumed until this point that the $p$-form fields we introduce are non-chiral. This condition was very important in our mathematical assumption that the total system of the gauge field with the $p$-form fields are given by a path integral over some configuration space $Q$. Chiral $p$-form fields, in contrast, do not have a simple path integral formulation in its physical spacetime dimensions $d$. Rather, it needs to be defined either abstractly by throwing away half of the degrees of freedom in dimension $d$, or more physically by considering a bulk Chern-Simons-like

theory in $d+1$ dimensions and considering its boundary modes on a $d$-dimensional boundary, see e.g. [18, 24–26]. Typically, one ends up constructing the theory of chiral $p$-form fields as an anomalous theory with a certain anomaly in $\Omega^{\text{spin}}_{d+1}(BG)$, which is then canceled against the fermion anomaly. This method is significantly different from the strategy used in this paper. It would be desirable to have a more uniform understanding of the Green-Schwarz cancellation using both chiral and non-chiral $p$-form fields.

So far, we only discussed the Green-Schwarz cancellation using continuous $p$-form fields. We can equally consider the Green-Schwarz cancellation using discrete $p$-form fields. Our general method can equally be applied to this case. Namely, if a discrete $p$-form $\mathbb{Z}_k$-valued field $H$ is to have a modified Bianchi identity schematically of the form $\delta H = c(F)$, where $c(F) \in H^{p+1}(BG; \mathbb{Z}_k)$ is the characteristic class to be trivialized, we represent $c$ as a map $c : BG \to K(\mathbb{Z}_k, p+1)$ between classifying spaces. Then its homotopy fiber $Q$ is a fibration of $K(\mathbb{Z}_k, p)$ over $BG$, and the rest of the analysis should go completely analogously. This might be useful when we expect the anomaly to be canceled by such a discrete $p$-form field, e.g. in the case discussed in Sec. 5 of [23].

In this context, we would like to note that, in [14, Sec. 5], a physics argument was already given that the 4d $SU(2)$ Witten anomaly, or more generally any global anomaly coming from the nontrivial gauge transformation in $\pi_d(G)$, cannot be canceled by topological degrees of freedom alone. Combined with the formulation of the anomaly cancellation using the homotopy fiber $Q$ given above, this implies a certain mathematical prediction that the bordism classes obtained from $\pi_d(G)$ should always lift to the bordism classes of $Q$. This can be easily checked, which is performed in Appendix C.

We listed above various possible extensions and generalizations of the considerations made in this paper. Even without generalizations or extensions, the methods developed in this paper should be useful in the analysis of the global anomaly cancellation by the introduction of $p$-form fields. The authors hope that some of the readers of this paper would use our methods to study such systems in the future.

## Acknowledgments

The authors would like to thank Kantaro Ohmori for illuminating discussions on the general issues treated in this paper, A. Debray, I. Basile, M. Delgado and M. Montero for correspondences on subtle technical issues in their paper [2] and also for the comments on the draft of this paper, and H. Zhang also for many suggestions to improve the draft of this paper. The authors also thank the anonymous referees for multiple constructive comments, which allowed the draft to be improved significantly.

**Funding information** SS is supported in part by Forefront Physics and Mathematics Program to Drive Transformation (FoPM), a World-leading Innovative Graduate Study (WINGS) Program, at the University of Tokyo. YT is supported in part by WPI Initiative, MEXT, Japan at Kavli IPMU, the University of Tokyo and by JSPS KAKENHI Grant-in-Aid (Kiban-C), No.24K06883.

## A More on the map $\Omega^{\text{spin}}_5(Q) \to \Omega^{\text{spin}}_5(BSp(n))$

In Sec. 3, we discussed the behavior of the fermion anomalies under the map $Q \to BSp(n)$; the crucial point was that the induced map $\Omega^{\text{spin}}_5(Q) \to \Omega^{\text{spin}}_5(BSp(n)) = \mathbb{Z}_2$ was an isomorphism. In the main text, this was shown by realizing a generator of $\Omega^{\text{spin}}_5(BSp(n))$ as a homotopically nontrivial map $S^5 \to BSp(n)$, and lifting it to $S^5 \to Q$ using the long exact sequence of homotopy groups. This was fairly straightforward.

In this appendix, we examine two different methods to show the same fact. They are both more complicated, but might be useful as an exercise in algebraic topology, and might possibly serve as a template for similar computations in a future related study.

## A.1 On $n$-equivalences and $(n-1)$-connectedness

We say that a map $f : X \to Y$ is an $n$-equivalence if the induced homomorphisms $\pi_k(X) \to \pi_k(Y)$ are isomorphic for $k < n$ and surjective for $k = n$. We say that a space $X$ is $(n-1)$-connected if $\pi_k(X) = 0$ for $k < n$, i.e. the map $X \to *$ is an $n$-equivalence. When $f : X \to Y$ is an $n$-equivalence, the long exact sequence of homotopy groups shows that the fiber $F(f)$ of $f$ is $(n-1)$-connected.

The homotopy excision theorem says the following. Let

$$
\begin{array}{ccc}
A & \xrightarrow{\ f\ } & X \\
\Big\downarrow{\scriptstyle g} & & \Big\downarrow{\scriptstyle g'} \\
Y & \xrightarrow{\ f'\ } & Z
\end{array}
\tag{A.1}
$$

be a homotopy pushout square. Assume $f$ is an $n$-equivalence and $g$ is an $m$-equivalence. Then the induced map of homotopy fibers $F(f) \to F(f')$ is an $(n+m-1)$-equivalence. For a proof, see e.g. [27, Theorem 9.3.5].

Let us list two corollaries of this theorem. Let us first take $Y = *$. Then $Z$ is the homotopy cofiber $C(f)$ of $f$. We find that, for an $(m-1)$-connected $A$ and an $n$-equivalence $f : A \to X$, the induced map $F(f) \to \Omega C(f)$ is an $(n+m-1)$-equivalence.

Let us further take $X = *$. Then $Z = \Sigma A$, and $F(f) = A$. We find that, for an $(m-1)$-connected $A$, the natural map $A \to \Omega\Sigma A$ is an $(2m-1)$-equivalence, i.e. $\pi_k(A) = \pi_{k+1}(\Sigma A)$ for $k < 2m-1$ and $\pi_{2m-1}(A) \to \pi_{2m}(\Sigma A)$ is surjective. This is the Freudenthal suspension theorem.

Let us now state how $(n-1)$-connectedness and $n$-equivalences affect the generalized homology groups. Let $G_*(-)$ be a connective generalized homology theory, i.e. those with $G_k(pt) = 0$ for $k < 0$. Then for an $(n-1)$-connected space $X$, we have $G_k(X) = G_k(pt)$ for $k < n$. This follows e.g. by first showing $H_k(X; \mathbb{Z}) = 0$ for $k < n$ by the Hurewicz theorem, and then using the Atiyah-Hirzebruch spectral sequence.

Next, consider an $n$-equivalence $f : X \to Y$. Then the induced map $G_k(X) \to G_k(Y)$ is an isomorphism for $k < n$. To see this, we compare the Atiyah-Hirzebruch spectral sequences associated to $F(f) \to X \to Y$ and to $* \to Y \to Y$. As $F(f)$ is $(n-1)$-connected, we have $G_k(F(f)) \simeq G_k(*)$ for $k < n$. The two spectral sequences then have the same $E^\infty_{p,q}$ in the range $p + q < n$. The sequences of extensions determining $G_k(-)$ from $E^\infty_{p,q}$ also behave naturally between the two spectral sequences, so that the induced map $G_k(X) \to G_k(Y)$ is an isomorphism for $k < n$.

Let us now suppose $A$ is $(m-1)$-connected and $f : A \to X$ is an $n$-equivalence. As we saw, $F(f) \to \Omega C(f)$ is an $(n+m-1)$-equivalence. As $F(f)$ is $(n-1)$-connected, $F(f) \to \Omega\Sigma F(f)$ is an $(2n-1)$-equivalence. Therefore, the natural map $\Sigma F(f) \to C(f)$ is an $\min(n+m, 2n)$-equivalence, and $\tilde{G}_{k-1}(F(f)) \simeq \tilde{G}_k(\Sigma F(f)) \to \tilde{G}_k(C(f))$ is an isomorphism for $k < \min(n+m, 2n)$. Using the fact that a cofiber sequence induces a long exact sequence in generalized homology, we conclude that there is a long exact sequence

$$
\begin{array}{rcccc}
 & & G_\ell(A) & \to & G_\ell(X) \\
\to & G_{\ell-1}(F(f)) & \to & G_{\ell-1}(A) & \to & G_{\ell-1}(X) \\
\to & G_{\ell-2}(F(f)) & \to & G_{\ell-2}(A) & \to & G_{\ell-2}(X) \\
\to & & \cdots,
\end{array}
\tag{A.2}
$$

where $\ell = \min(n+m, 2n) - 1$.

## A.2 Adams spectral sequence for $\Omega_5^{\text{spin}}(Q) \to \Omega_5^{\text{spin}}(BSp(n))$

We now use the consideration above by taking $A = BSp(n)$, $X = K(\mathbb{Z}, 4)$, and $f = q_1 : BSp(n) \to K(\mathbb{Z}, 4)$. Then $Q = F(f)$ and we have $m = 4$ and $n = 5$. We can then use (A.2) to study the map $\Omega_5^{\text{spin}}(Q) \to \Omega_5^{\text{spin}}(BSp(n))$.

More specifically, we first take $G_*(-) = H_*(-;\mathbb{Z}_2)$ to study the relation among $H^*(Q;\mathbb{Z}_2)$, $H^*(BSp(n);\mathbb{Z}_2)$, and $H^*(K(\mathbb{Z},4);\mathbb{Z}_2)$ as $\mathcal{A}(1)$ modules. Since the long exact sequence (A.2) is valid for at least values up to 7, the following three isomorphisms hold:

$$H_7(K(\mathbb{Z},4);\mathbb{Z}_2) \longrightarrow H_7(\Sigma Q;\mathbb{Z}_2), \tag{A.3}$$

$$H_6(K(\mathbb{Z},4);\mathbb{Z}_2) \longrightarrow H_6(\Sigma Q;\mathbb{Z}_2), \tag{A.4}$$

$$H_4(BSp(n);\mathbb{Z}_2) \longrightarrow H_4(K(\mathbb{Z},4);\mathbb{Z}_2), \tag{A.5}$$

where each group corresponds to $\mathbb{Z}_2$, and note that we use the fact that $H_{n-1}(Q;\mathbb{Z}_2) \simeq H_n(\Sigma Q;\mathbb{Z}_2)$. Note also that any unmentioned group is zero in this range.

As the second step we take $G_* = \Omega_*^{\text{spin}}$ and use the Adams spectral sequence, utilizing the information obtained in the first step. The above homological isomorphisms give rise to the short exact sequences in cohomology and as $\mathcal{A}(1)$ modules up to degree 7:

$$0 \leftarrow \widetilde{H^*}(BSp(n);\mathbb{Z}_2)_{\le 7} \leftarrow \widetilde{H^*}(K(\mathbb{Z},4);\mathbb{Z}_2)_{\le 7} \leftarrow \widetilde{H^*}(\Sigma Q;\mathbb{Z}_2)_{\le 7} \leftarrow 0, \tag{A.6}$$

where the direction of the arrows is reversed because we consider the cohomology. The cohomological short exact sequence, when explicitly showing the action of $\mathcal{A}(1)$, takes the following form:

$$\begin{array}{c}
\text{(diagram)}
\end{array} \tag{A.7}$$

Thus, we obtain the long exact sequence of the $E_2$ page of the Adams spectral sequence [1]. The map of the long exact sequence that we want can be seen in the following Adams chart, specifically it is the lower of the two arrows:

$$\begin{array}{c}
\text{(Adams chart)}
\end{array} \tag{A.8}$$

Note that there are no nontrivial differentials in the range we are focusing on, which means that the $E_2$ page has already converged. Since $\text{Ext}_{\mathcal{A}(1)}^{*,6}(K(\mathbb{Z},4),\mathbb{Z}_2) = 0$, we obtain the isomorphism:

$$\text{Ext}_{\mathcal{A}(1)}^{0,6}(\Sigma Q, \mathbb{Z}_2) \xrightarrow{\simeq} \text{Ext}_{\mathcal{A}(1)}^{1,6}(BSp(n), \mathbb{Z}_2). \tag{A.9}$$

At the bordism level, this isomorphism induces $\Omega_6^{\text{Spin}}(\Sigma Q) \to \Omega_5^{\text{Spin}}(BSp(n))$. Considering (A.2) again, the isomorphism $\Omega_6^{\text{Spin}}(\Sigma Q) \simeq \Omega_5^{\text{Spin}}(Q)$ follows, and we finally confirm that $\Omega_5^{\text{Spin}}(Q) \to \Omega_5^{\text{Spin}}(BSp(n))$ behaves as an identity.

## A.3 Atiyah-Hirzebruch spectral sequence for $\Omega_5^{\mathrm{spin}}(Q) \to \Omega_5^{\mathrm{spin}}(BSp(n))$

We can also use the Atiyah-Hirzebruch spectral sequence to study the map $\Omega_5^{\mathrm{spin}}(Q) \to \Omega_5^{\mathrm{spin}}(BSp(n))$. Recall the integral cohomology group of $Q$ determined in Sec. 5.2. Feeding this info to the Atiyah-Hirzebruch spectral sequence for $\Omega_\bullet^{\mathrm{spin}}$, we see that three bottom lines of $E^2$ page are

$$
\begin{array}{c}
\\
K(\mathbb{Z},3)
\end{array}
\begin{array}{c|ccccccc}
2 & \mathbb{Z}_2 & 0 & 0 & \mathbb{Z}_2 & 0 & \mathbb{Z}_2 & \mathbb{Z}_2 \\
1 & \mathbb{Z}_2 & 0 & 0 & \mathbb{Z}_2 & 0 & \mathbb{Z}_2 & \mathbb{Z}_2 \\
0 & \mathbb{Z} & 0 & 0 & \mathbb{Z} & 0 & \mathbb{Z}_2 & 0 \\
\hline
& 0 & 1 & 2 & 3 & 4 & 5 & 6
\end{array}
$$

$$\downarrow$$

$$
\begin{array}{c}
\\
Q
\end{array}
\begin{array}{c|ccccccc}
2 & \mathbb{Z}_2 & 0 & 0 & 0 & 0 & \mathbb{Z}_2 & \mathbb{Z}_2 \\
1 & \mathbb{Z}_2 & 0 & 0 & 0 & 0 & \mathbb{Z}_2 & \mathbb{Z}_2 \\
0 & \mathbb{Z} & 0 & 0 & 0 & 0 & \mathbb{Z}_2 & 0 \\
\hline
& 0 & 1 & 2 & 3 & 4 & 5 & 6
\end{array}
\quad , \qquad (A.10)
$$

$$\downarrow$$

$$
\begin{array}{c}
\\
BSp(n)
\end{array}
\begin{array}{c|ccccccc}
2 & \mathbb{Z}_2 & 0 & 0 & 0 & \mathbb{Z}_2 & 0 & 0 \\
1 & \mathbb{Z}_2 & 0 & 0 & 0 & \mathbb{Z}_2 & 0 & 0 \\
0 & \mathbb{Z} & 0 & 0 & 0 & \mathbb{Z} & 0 & 0 \\
\hline
& 0 & 1 & 2 & 3 & 4 & 5 & 6
\end{array}
$$

where the shaded pair is killed, because $H_5(Q,\mathbb{Z})$ is detected by its mod-2 reduction paired with $\mathrm{Sq}^2 u$ where $u$ is the mod-2 reduction of the generator of $H^3(K(\mathbb{Z},3),\mathbb{Z}) \simeq \mathbb{Z}$.

In this manner, we have

$$
\Omega_5^{\mathrm{spin}}(K(\mathbb{Z},3))_{=0} \to \Omega_5^{\mathrm{spin}}(Q)_{=\mathbb{Z}_2} \xrightarrow{f} \Omega_5^{\mathrm{spin}}(BSU(2))_{=\mathbb{Z}_2}, \qquad (A.11)
$$

and the question is if the second arrow $f$ is an identity or a zero map. We can try to obtain more information by using the functoriality of Atiyah-Hirzebruch spectral sequence.[14] We have

$$
\begin{array}{c|ccccccccc}
& 0 \to & E_{4,1} & \to & \Omega_5^{\mathrm{spin}} & \to & E_{5,0} & \to 0 \\
\hline
Q & 0 \to & 0 & \to & \mathbb{Z}_2 & \to & \mathbb{Z}_2 & \to 0 \\
& & \downarrow & & \downarrow f & & \downarrow & \\
BSU(2) & 0 \to & \mathbb{Z}_2 & \to & \mathbb{Z}_2 & \to & 0 & \to 0
\end{array}
\quad . \qquad (A.12)
$$

Both the zero map and the identity are compatible at this stage.[15] We know that the map $f$ is an isomorphism from a different argument in Sec. 3, and we cannot directly see that if we used the Atiyah-Hirzebruch spectral sequence in this manner.

A way out is to compare two Atiyah-Hirzebruch spectral sequences for

$$
H_p(BSp(n), \Omega_q^{\mathrm{spin}}(K(\mathbb{Z},3))) \Rightarrow \Omega_{p+q}^{\mathrm{spin}}(Q), \qquad (A.13)
$$

and

$$
H_p(BSp(n), \Omega_q^{\mathrm{spin}}(pt)) \Rightarrow \Omega_{p+q}^{\mathrm{spin}}(BSp(n)). \qquad (A.14)
$$

---

[14]The authors thank Kantaro Ohmori for explanation.

[15]If it had been

$$
\begin{array}{ccccccc}
0 \to & \mathbb{Z}_2 & \to & \mathbb{Z}_2 & \to & 0 & \to 0 \\
& \downarrow & & \downarrow g & & \downarrow & \\
0 \to & 0 & \to & \mathbb{Z}_2 & \to & \mathbb{Z}_2 & \to 0
\end{array}
\quad ,
$$

$g$ would have been forced to zero.

Using $\Omega_q^{\mathrm{spin}}(K(\mathbb{Z},3))$ which can be readily computed from (A.10), we have

$$
Q \quad
\begin{array}{c|ccccccc}
6 & 0 \\
5 & 0 & 0 \\
4 & \mathbb{Z} & 0 & 0 \\
3 & \boxed{\mathbb{Z}} & 0 & 0 & 0 \\
2 & \mathbb{Z}_2 & 0 & 0 & 0 & \mathbb{Z}_2 \\
1 & \mathbb{Z}_2 & 0 & 0 & 0 & \mathbb{Z}_2 & 0 \\
0 & \mathbb{Z} & 0 & 0 & 0 & \boxed{\mathbb{Z}} & 0 & 0 \\
\hline
& 0 & 1 & 2 & 3 & 4 & 5 & 6
\end{array}
$$

$$\downarrow$$

$$
BSp(n) \quad
\begin{array}{c|ccccccc}
6 & 0 \\
5 & 0 & 0 \\
4 & \mathbb{Z} & 0 & 0 \\
3 & 0 & 0 & 0 & 0 \\
2 & \mathbb{Z}_2 & 0 & 0 & 0 & \mathbb{Z}_2 \\
1 & \mathbb{Z}_2 & 0 & 0 & 0 & \mathbb{Z}_2 & 0 \\
0 & \mathbb{Z} & 0 & 0 & 0 & \mathbb{Z} & 0 & 0 \\
\hline
& 0 & 1 & 2 & 3 & 4 & 5 & 6
\end{array}
$$

(A.15)

at the level of the $E^2$ page. Here, the shaded pair is killed, and the $\mathbb{Z}_2$ in $\Omega_5^{spin}(Q)$ and $\Omega_5^{spin}(BSp(n))$ both come from the same entry $E_{4,1}^2$ of the $E^2$ page of the respective Atiyah-Hirzebruch spectral sequences. Therefore $f$ is seen to be an identity.

# B   The computation of $\pi_4(S^3)$

Here we review the classic computation of $\pi_4(S^3) = \mathbb{Z}_2$ by using the Leray-Serre spectral sequence, which can be found in any basic textbook of algebraic topology. We include this appendix in this paper, since the arguments which will be employed here will be perfectly analogous to our arguments in Sec. 3 and Sec. 5.2. Although we can replace $S^3$ by $Sp(n)$ for general $n$ below without any change, we stick to the case $Sp(1) = S^3$ for notational simplicity.

Let us consider the classifying map $f : S^3 \to K(\mathbb{Z},3)$ of the generator of $H^3(S^3,\mathbb{Z}) \simeq \mathbb{Z}$, and $P$ be its homotopy fiber:

$$P \to S^3 \to K(\mathbb{Z},3). \tag{B.1}$$

Correspondingly, we have a fibration of the form

$$K(\mathbb{Z},2) \to P \to S^3. \tag{B.2}$$

The homotopy long exact sequence associated to (B.1) is

$$
\begin{array}{rcccl}
& & & \to & \pi_5(K(\mathbb{Z},3))_{=0} \\
\to & \pi_4(P) & \to & \pi_4(S^3)_{=?} & \to & \pi_4(K(\mathbb{Z},3))_{=0} \\
\to & \pi_3(P) & \to & \pi_3(S^3)_{=\mathbb{Z}} & \xrightarrow{\sim} & \pi_3(K(\mathbb{Z},3))_{=\mathbb{Z}} \\
\to & \pi_2(P) & \to & \pi_2(S^3)_{=0} & \to & \pi_2(K(\mathbb{Z},3))_{=0} \\
\to & \pi_1(P) & \to & \pi_1(S^3)_{=0} & \to & \pi_1(K(\mathbb{Z},3))_{=0} \\
\to & \pi_0(P) & \to & \pi_0(S^3)_{=0} & \to & \pi_0(K(\mathbb{Z},3))_{=0}
\end{array}
, \tag{B.3}
$$

where $\pi_3(S^3) \simeq \pi_3(K(\mathbb{Z},3))$ is by construction. From this we know

$$\pi_{0,1,2,3}(P) = 0, \qquad \pi_4(P) \simeq \pi_4(S^3), \tag{B.4}$$

and an application of Hurewicz theorem gives

$$H_0(P;\mathbb{Z}) = \mathbb{Z}, \qquad H_{1,2,3}(P;\mathbb{Z}) = 0, \qquad H_4(P;\mathbb{Z}) \simeq \pi_4(P) \simeq \pi_4(S^3). \tag{B.5}$$

Therefore it suffices to compute $H_4(P,\mathbb{Z})$.

For this we use the Leray-Serre spectral sequence for $H^\bullet(P;\mathbb{Z})$, associated to $K(\mathbb{Z},2) \to P \to S^3$. Recall $K(\mathbb{Z},2) = \mathbb{CP}^\infty$ and $H^*(K(\mathbb{Z},2);\mathbb{Z}) = \mathbb{Z}[u]$ where $u$ is of degree 2. Let us denote the generator of $H^3(S^3;\mathbb{Z}) = \mathbb{Z}$ by $v$. Then the $E_2$ page is simply

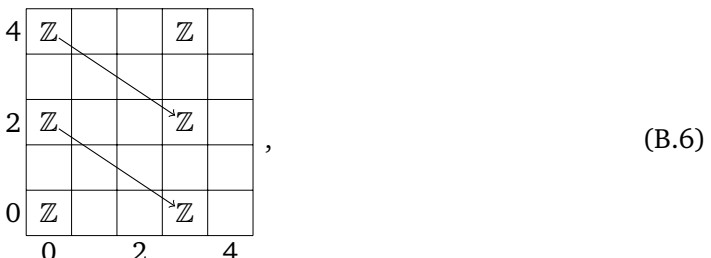

$$\tag{B.6}$$

where $E_2^{0,2n} \simeq \mathbb{Z}u^n$ and $E_2^{3,2n} \simeq \mathbb{Z}u^n v$.

To determine $d_2$, note that we have $H^2(P;\mathbb{Z}) = H^3(P;\mathbb{Z}) = 0$ from (B.5). Therefore $d_2 : E_2^{0,2} \to E_2^{3,0}$ is an isomorphism given by $d_2 u = v$. Then $d_2 : E_2^{0,4} \to E_2^{3,2}$ is a multiplication by 2, given by $d_2 u^2 = 2uv$. Therefore the $E_3$ page is

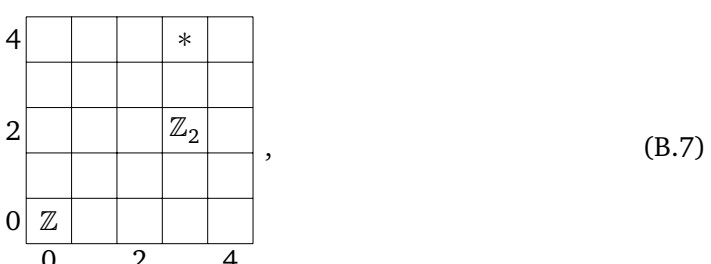

$$\tag{B.7}$$

from which we conclude

$$H^4(P;\mathbb{Z}) = 0, \qquad H^5(P;\mathbb{Z}) = \mathbb{Z}_2. \tag{B.8}$$

Therefore we have

$$H_4(P;\mathbb{Z}) = \mathbb{Z}_2, \tag{B.9}$$

and by (B.5) we conclude

$$\pi_4(S^3) = \mathbb{Z}_2. \tag{B.10}$$

## C  Traditional global anomalies and the Green-Schwarz mechanism

In this appendix we generalize our argument in Sec. 3 to show that it is impossible to cancel traditional global anomalies associated to $\pi_d(G)$ in terms of the Green-Schwarz mechanism using non-chiral $p$-form fields, either discrete or continuous.[16] For discrete $p$-form fields, this conclusion is compatible with the result derived in a different way in [14, Sec. 5] that traditional global anomalies associated to $\pi_d(G)$ cannot be canceled by topological degrees of freedom.

---

[16]The authors thank the anonymous referee who asked a question leading to the result in this appendix.

We are interested in the anomaly detected by the torsion element in the image of the map (6), namely,

$$\pi_d(G) \simeq \pi_{d+1}(BG) \xrightarrow{f} \Omega^{\text{spin}}_{d+1}(BG). \tag{C.1}$$

The introduction of a $p$-form field replaces the configuration space $BG$ by some fibration $Q$ over $BG$, whose fiber is $K(A, p)$ where $A = \mathbb{Z}$ for a continuous $p$-form field and $A = \mathbb{Z}_k$ for a discrete $p$-form field.

We are interested whether the image of $f$ lifts under the map

$$\Omega^{\text{spin}}_{d+1}(Q) \xrightarrow{g} \Omega^{\text{spin}}_{d+1}(BG). \tag{C.2}$$

To analyze this, we use the following commutative diagram:

$$
\begin{array}{ccc}
\pi_{d+1}(Q) & \xrightarrow{\tilde{f}} & \Omega^{\text{spin}}_{d+1}(Q) \\
\downarrow{\scriptstyle\tilde{g}} & & \downarrow{\scriptstyle g} \\
\pi_{d+1}(BG) & \xrightarrow{f} & \Omega^{\text{spin}}_{d+1}(BG)
\end{array}
\quad . \tag{C.3}
$$

So it suffices to show that $\tilde{g}$ is a surjection, under a suitable condition on $p$.

For this, we use the long exact sequence of homotopy groups associated to the fibration $K(A, p) \to Q \to BG$. The relevant part of the sequence is

$$\pi_{d+1}(K(A, p)) \to \pi_{d+1}(Q) \to \pi_{d+1}(BG) \to \pi_d(K(A, p)). \tag{C.4}$$

For a non-zero element of $\pi_{d+1}(BG)$ not to lift to $\pi_{d+1}(Q)$, its image in $\pi_d(K(A, p))$ must be non-zero. This requires $p = d$, in which case $\pi_d(K(A, p)) \simeq A$. As the element of $\pi_{d+1}(BG)$ in question is assumed to be torsion, $A$ must have a torsion element, which requires $A$ to be a finite Abelian group. This means that we are introducing a discrete $d$-form field $C_d$ with the coupling $\int_{M_d} C_d$, whose gauge variation cancels the global anomaly, almost by construction. But this coupling make the partition function zero, due to the path integral over the gauge field $C_d$, which is $\sum_{k=1}^{N} e^{2\pi i k/N} = 0$ for $A = \mathbb{Z}_N$. This is the only possibility, but we do not usually regard this as the anomaly cancellation. This was what we wanted to show.

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
