# Peer review of "Cancelling mod-2 anomalies by Green-Schwarz mechanism with $B_{μν}$"

_SciPost Physics, doi:SciPost Phys. 19, 017 (2025)_

## Round 2 · Referee Report · Anonymous (Referee 1) · 2025-2-13

Report
The essential part of the analysis consists of introducing a B-field implementing a Green-Schwarz mechanism for local anomaly cancellation, and then developing a framework to study global anomalies in the presence of the B-field. The B-field has anomalous Bianchi identities, which have to be encoded in the structure of the anomaly.
In order to do so, the authors follow a strategy based on homotopy fibers which can eliminate the characteristic class in the Bianchi identity. This alters the configuration space of anomaly backgrounds. The authors then provide a detailed spectral-sequence computation of the relevant bordism groups which capture the anomaly. As a result, while the four-dimensional case cannot be canceled in this fashion, the eight-dimensional ones can. The authors point out that this is consistent with the existence of string-theoretic realizations of this theory, for which the anomaly does indeed cancel. The key result is the vanishing of the map in eq. (4.12),
The paper is well-written, clear and concise. Several details are given to reproduce computations, and the problem at stake is motivated and studied step by step, thoroughly and with precision. The results are of significance both for applications in quantum field theory from a bottom-up perspective and for string theory in a top-down sense. In the latter context, the cancellation of the eight-dimensional anomaly comes from a detailed balance of ingredients, which provides yet another "miracle" as the authors point out. The appendices are also useful for the reader, providing substantial material which complements the main text.
My only suggestion for the authors is to provide a concluding outlook to connect the result to a broader research context, although the paper is already fine without it.
All in all, I am glad to recommend this paper for publication on SciPost.
Requested changes
1-I noticed the typo "such that the relation (...) are satisfied" right above eq. (1.1).
2-I would suggest the authors to include a concluding outlook, briefly discussing the implications of their results in relation to other applications and interesting cases to study. For instance, the role played by twisted string structures or the possibility of additional anomalies in the presence of gravity.
Recommendation
Publish (easily meets expectations and criteria for this Journal; among top 50%)

---

## Round 2 · Referee Report · Anonymous (Referee 2) · 2025-2-19

Report
This paper derives a necessary and sufficient condition for determining whether a particular fermion global anomaly (for symmetry $G$) can be cancelled by a Green-Schwarz mechanism involving a background $p$-form gauge field $H_p$, with modified Bianchi identity relating $dH_p$ to a characteristic class $X$ built from the $G$-gauge and tangent bundles. The key idea is to formulate the configuration space for the background fields as the homotopy fibre $Q$ of a map specified by $X$. This $Q$ is itself realised as a fibration over $BG$. From here, forgetful maps from the bordism groups of $Q$ to the bordism groups of $BG$ can be established, with which one can ascertain whether global anomalies in $G$ remain non-trivial under pulling back to the $Q$ structure. The condition, which supersedes previous conditions in the literature, is demonstrated (through two examples) to be powerful: the physics question formulated above is reduced to determining whether a particular map between bordism groups is zero or not. I think this is an excellent paper, containing many new mathematical computations of bordism groups and maps between that will surely be useful to researchers in this field, and which provides a rigorous answer to a well-posed question about anomaly cancellation. I happily recommend this paper for publication in SciPost.
I have a few minor comments and questions regarding the manuscript and the scope of the central result, which I would be grateful if the authors addressed: - Both examples feature $\mathbb{Z}_2$-valued chiral fermion anomalies, and indeed a restriction to 'mod-2 anomalies' is even made in the title of the paper. While I agree that the mod-2 anomalies are the most relevant examples to study, nothing in the formalism appears specific to mod 2 anomalies. I wonder whether the same criterion applies for any ($\mathbb{Z}_k$-valued) global anomaly? - Related to this, the $p$-form gauge field is assumed to be continuous, with classifying space (in isolation) given by $K(\mathbb{Z},p+1)$. It is then investigated whether this can cancel global mod 2 fermion anomalies. Can one consider also discrete $p$-form gauge fields, and/or modified Bianchi identities that involve torsion classes? Is it possible to show, for example, that even with these more general ingredients one still cannot cancel the 4d $SU(2)$ Witten anomaly? - I am curious as to whether there is any connection to the non-perturbative formulation of Green-Schwarz terms via shifted Wu-Chern-Simons (WCS) terms, as set out e.g. by Monnier and Moore [1808.01334], following Monnier [1607.01396]. I recall these WCS actions playing a role in similar studies of global anomalies in 6d. - In section 3.1, the authors consider the 4d $SU(2)$ anomaly via a particular representative $f:S^5 \to BSU(2)$ of $\Omega_5^\text{Spin}(BSU(2))$ that is different to the two representatives that were described previously in Section 2.1. So it remains a little cryptic how this bundle over $S^5$ is constructed; for me, a sentence or two of explanation would help here. One way to describe this $f$ is via the clutching construction: take an $S^4$ with a zero-instanton configuration, which can be extended to a hemisphere $D^4$, and glue it to an equal and oppositely-oriented hemisphere using a non-trivial gauge transformation $g:S^4\to SU(2)$, characterized by a homotopy class $[g]\in \pi_4(SU(2))=\mathbb{Z}_2$. The clutching construction tells us that $[g]$ equals the homotopy class of the resulting bundle over the $S^5$. This underlies the authors' remark that the bundle 'comes from a nontrivial element of $\pi_5(BSU(2))=\pi_4(SU(2))=\mathbb{Z}_2$'. - Out of curiosity, I wonder if there is any particular outlook for using these results? Can the authors think of other interesting examples which this methodology can be applied to, namely certain global anomalies that we do not know can be cancelled via Green-Schwarz?
Requested changes
The key final sentence of section 2.3 is confusingly written: should it not rather be 'Taking the contrapositive, the condition $p^\ast(I)=0$ for a global anomaly {\em to be cancelled by the Green-Schwarz mechanism} is that ...'
Recommendation
Publish (surpasses expectations and criteria for this Journal; among top 10%)

---

## Round 3 · Author Response

We are very grateful to the referees for constructive comments. We list our replies to the referees' comments and the improvements made to the draft. We hope that the article is now acceptable for publication.

Reply to Referee #1:

  1. The typo was corrected by replacing "the relation" by "the relations".

  2. We added a concluding section.

Reply to Referee #2:

The referee asked us a few questions within the "Report" section. Let us first respond to these.

  • The first question was whether our method is applicable to $\mathbb{Z}_k$ anomaly for $k\neq 2$. Indeed, our method does not require that the anomaly is mod 2. The same analysis applies equally well to other mod k anomalies, and in fact to the perturbative anomalies as well, since the bordism formulation of anomalies is very general. We added a few comments on this issue in the new Sec. 6.

  • The second question was whether we can treat the effects of discrete $p$-form gauge fields on the anomalies in the same manner. Indeed, our method is general in that we can analyze the effect of the introduction of discrete $p$-form fields to the anomaly by constructing a fibration $Q$ of $K(\mathbb{Z}_k,p)$ over $BG$ and studying the bordism group of $Q$. A few comments on this issue is now in Sec. 6.

  • The referee then asked whether the introduction of discrete $p$-form gauge fields can cancel Witten's $SU(2)$ anomaly in 4d. This is an interesting question. There is a physics argument saying that Witten's SU(2) anomaly can never be canceled without additional massless fields, given in Sec. 5 of https://arxiv.org/abs/1710.04218 . This translates to mathematical predictions concerning bordism groups of fibrations $Q$ of $K(\mathbb{Z}_k,p)$ over $BSU(2)$ never vanishes. It is not hard to confirm this prediction. We added a new Appendix C to explain this point; we would like to thank the referee for asking this very interesting question.

  • The referee's next question was about the relation to the shifted Wu structure and Wu-Chern-Simons theories used by Monnier, Moore and collaborators. We should mention that Monnier et al. used the shifted Wu structure and the Wu-Chern-Simons theories to formulate chiral $p$-form fields, whereas our intention in this paper is to study non-chiral $p$-form fields. As chiral $p$-form fields cannot be formulated by a path integral over a modified configuration space, the method we developed in the paper does not apply. Therefore, we say that there definitely is a relation, but the relation is distant. Again we made some comments in Sec. 6.

  • The next comment was that our description of the generator of $\Omega^\text{spin}_5(BSU(2))=\mathbb{Z}_2$ at the beginning of Sec. 3 was too terse and also confused. We agree with the referee's assessment here. We added more discussions in Sec.2.1 when these bordism classes were first treated, and we referred to Sec.2.1 from Sec.3.

  • The final question within the main "Report" section was the outlook for using the methods described in this paper to other situations within string theory or hep-th in general. This is addressed in the new concluding section, Sec. 6.

The referee also requested to change another confusing sentence at the end of Sec.2.3. This was corrected as suggested by the referee.

---

## Round 3 · List of Changes

(It's incorporated to the reply above)

---

## Editorial Decision

published